# Circulating Liquid Biopsy Biomarkers in Glioblastoma: Advances and Challenges

**DOI:** 10.3390/ijms25147974

**Published:** 2024-07-21

**Authors:** Attila A. Seyhan

**Affiliations:** 1Laboratory of Translational Oncology and Experimental Cancer Therapeutics, Warren Alpert Medical School, Brown University, Providence, RI 02912, USA; attila_seyhan@brown.edu; 2Department of Pathology and Laboratory Medicine, Warren Alpert Medical School, Brown University, Providence, RI 02912, USA; 3Joint Program in Cancer Biology, Lifespan Health System and Brown University, Providence, RI 02912, USA; 4Legorreta Cancer Center, Brown University, Providence, RI 02912, USA

**Keywords:** glioblastoma, biomarkers, diagnosis, prognosis, cell-free DNA (cfDNA), circulating tumor DNA (ctDNA), circulating microRNAs (miRNAs), circulating tumor cells (CTCs), extracellular vesicles (EVs) and exosomes, proteomics, metabolomics

## Abstract

Gliomas, particularly glioblastoma (GBM), represent the most prevalent and aggressive tumors of the central nervous system (CNS). Despite recent treatment advancements, patient survival rates remain low. The diagnosis of GBM traditionally relies on neuroimaging methods such as magnetic resonance imaging (MRI) or computed tomography (CT) scans and postoperative confirmation via histopathological and molecular analysis. Imaging techniques struggle to differentiate between tumor progression and treatment-related changes, leading to potential misinterpretation and treatment delays. Similarly, tissue biopsies, while informative, are invasive and not suitable for monitoring ongoing treatments. These challenges have led to the emergence of liquid biopsy, particularly through blood samples, as a promising alternative for GBM diagnosis and monitoring. Presently, blood and cerebrospinal fluid (CSF) sampling offers a minimally invasive means of obtaining tumor-related information to guide therapy. The idea that blood or any biofluid tests can be used to screen many cancer types has huge potential. Tumors release various components into the bloodstream or other biofluids, including cell-free nucleic acids such as microRNAs (miRNAs), circulating tumor DNA (ctDNA), circulating tumor cells (CTCs), proteins, extracellular vesicles (EVs) or exosomes, metabolites, and other factors. These factors have been shown to cross the blood-brain barrier (BBB), presenting an opportunity for the minimally invasive monitoring of GBM as well as for the real-time assessment of distinct genetic, epigenetic, transcriptomic, proteomic, and metabolomic changes associated with brain tumors. Despite their potential, the clinical utility of liquid biopsy-based circulating biomarkers is somewhat constrained by limitations such as the absence of standardized methodologies for blood or CSF collection, analyte extraction, analysis methods, and small cohort sizes. Additionally, tissue biopsies offer more precise insights into tumor morphology and the microenvironment. Therefore, the objective of a liquid biopsy should be to complement and enhance the diagnostic accuracy and monitoring of GBM patients by providing additional information alongside traditional tissue biopsies. Moreover, utilizing a combination of diverse biomarker types may enhance clinical effectiveness compared to solely relying on one biomarker category, potentially improving diagnostic sensitivity and specificity and addressing some of the existing limitations associated with liquid biomarkers for GBM. This review presents an overview of the latest research on circulating biomarkers found in GBM blood or CSF samples, discusses their potential as diagnostic, predictive, and prognostic indicators, and discusses associated challenges and future perspectives.

## 1. Introduction

Tumors of the central nervous system (CNS), particularly high-grade gliomas like glioblastoma (GBM), are the most prevalent and aggressive primary malignant tumors in the CNS among adults and present significant challenges due to their aggressiveness and poor prognosis [1,2].

GBM accounts for 14.2% of all diagnosed CNS tumors and 50.1% of all malignant tumors in the USA, with a median survival time of about 15 months, regardless of treatment, showing little improvement despite extensive research [3]. The annual incidence (or number of new cases) of new GBM cases in the USA is 3.19 per 100,000 people, with a prevalence (number of existing cases) of 9.23 per 100,000 people [4]. Over 14,490 US residents are expected to receive a GBM diagnosis in 2023. International studies show an incidence rate of GBM ranging from 0.59 to 5 per 100,000 persons, with an increasing trend in many countries [5]. The incidence is 1.6 times higher in males than in females and 2.0 times higher in Caucasians compared to in Africans and Afro-Americans, with lower rates in Asians and American Indians [6]. Factors contributing to this increase include aging populations, ionizing radiation, air pollution, and overdiagnosis [5], as well as smoking, pesticides, and certain occupations [7]. Additional, albeit not established, connections include viral infections such as simian virus 40 (SV40) [8], human herpesvirus 6 (HHV-6) [9,10], and cytomegalovirus (CMV) [11,12,13]. However, the viral hypothesis regarding the etiology of GBM remains unestablished and is considered controversial by many experts.

The 2021 WHO classification introduced significant revisions to CNS tumor classification, integrating molecular parameters with histology to define various tumor entities [14]. This approach redefined various tumor entities, including different subtypes of diffuse gliomas (e.g., glioneuronal and neuronal tumors), choroid plexus tumors, embryonal tumors, pineal tumors, cranial and paraspinal nerve tumors, meningiomas, mesenchymal, non-meningothelial tumors involving the CNS, melanocytic tumors, hematolymphoid tumors involving the CNS, germ cell tumors, tumors of the sellar region, metastases to the CNS, and genetic tumor syndromes involving the CNS, and introduced new entities distinguished by both histological and molecular characteristics [14,15,16].

These include various subtypes of GBM such as isocitrate dehydrogenase (*IDH*)-wildtype and *IDH*-mutant; diffuse midline glioma, H3 K27M-mutant; RELA fusion-positive ependymoma; medulloblastoma, WNT-activated and medulloblastoma, SHH-activated; and embryonal tumor with multilayered rosettes, C19MC-altered. Gliomas are further classified according to their cellular origins, such as oligodendrogliomas, which arise from oligodendrocytes, ependymomas, which arise from ependymal cells, and astrocytomas, which arise from astrocytes [17]. Astrocytomas, further categorized by WHO definitions according to the malignancy grade (ranging from I to IV), include GBMs [17], the most common and lethal form.

Genome, transcriptome, and proteome profiling has identified three subtypes of GBMs: proneural, classic, and mesenchymal [18,19,20,21], each exhibiting distinct genetic alterations affecting treatment responses and patient prognosis [18]. These studies indicate that GBM tumors exhibit distinct molecular features such as telomerase reverse transcriptase (*TERT*) promoter mutation, epidermal growth factor receptor (*EGFR*) gene amplification, the combined gain of the entire chromosome 7, and the loss of the entire chromosome 10 [14,22]. In addition, GBM tumors exhibit distinct structural features such as high cellular and microvascular proliferation, tumor infiltration, and core necrosis.

Despite advancements, GBM patients typically succumb to the disease within two years of diagnosis [22,23,24,25], with a median survival of less than 15 months and a 5-year survival rate of only 6.8% [26]. Only 10% of patients respond to standard-of-care (SoC) therapies, highlighting the urgent need for more effective treatments [27,28]. The current standard-of-care for GBM includes surgical resection, radiotherapy, and chemotherapy with temozolomide (TMZ) as the primary chemotherapeutic agent [29,30]. TMZ’s efficacy depends on the methylation status of the O6-methylguanine-DNA methyltransferase (*MGMT*) promoter, which can increase tumor cell sensitivity to its DNA-damaging effects [31]. Despite these efforts, patients treated with TMZ have a median survival of around 15 months [26], partly due to therapy resistance and high relapse rates [30,32,33,34,35].

Recent research suggests that GBM cancer stem cells (GSCs), which are resistant to radiation and chemotherapy, may also contribute to rapid tumor recurrence [32,33,34,35]. Novel therapies, such as immunotherapies (IT), including anti-programmed cell death protein 1 (PD-1) immune checkpoint inhibitor (ICI) and nivolumab, and drugs targeting vascular endothelial growth factor (VEGF), such as bevacizumab, are being explored, but their efficacy in GBM treatment remains mixed [36,37,38] due to the tumor’s low mutational burden and immunologically cold nature [39]. Still, combined therapies with traditional checkpoints such as PD-1/PD-L1, CTLA-4, TIM-3, and others may offer benefits by altering the tumor microenvironment (TME) [36,40,41]. Traditional anti-PD-1 therapy may also be combined with other targets, such as TIM-3 and BTLA. There is also interest in combining immunotherapy with another targetable mechanism. The interest in anti-CD276 studies combined with bevacizumab arises from the known connection between CD276/B7-H3 and angiogenesis [41].

Current GBM diagnosis relies on neuroimaging such as MRI or CT scans [29] followed by either surgical resection or tissue biopsies of the tumor tissue to confirm the diagnosis, determine its grade, and characterize its properties. However, neuroimaging techniques such as MRI or CT scans may not reliably differentiate tumor progression from treatment-related changes [42]. Additionally, the serial collection of tissue biopsies to monitor dynamic changes in the tumor throughout the therapy period may not be feasible.

Because tumors, including GBM, generally release tumor content into both the bloodstream and [43] cerebrospinal fluid (CSF) [44], liquid biopsies offer a minimally invasive or non-invasive alternative for longitudinally measuring circulating biomarkers. Currently, various types of circulating biomarkers are being explored, including circulating tumor DNA (ctDNA), micro RNAs (miRNAs), circulating tumor cells, (CTCs), extracellular vesicles (EVs) and exosomes, proteins and metabolites in serially collected blood, CSF, and other biofluids, to monitor dynamic changes in the tumor throughout the therapy period [42,45,46].

Note that since these circulating biomarkers are not yet approved through formal regulatory processes, they cannot currently replace standard risk stratification methods in routine clinical diagnostics. However, once they receive regulatory approval, they could complement standard diagnostic and risk stratification methods. This would allow for the real-time and dynamic monitoring of tumor characteristics and treatment responses, as well as the prediction of disease prognosis through the serial sampling of GBM patients.

In summary, this review aims to provide an overview of the current literature on circulating biomarkers as potential minimally invasive or non-invasive tools for guiding the treatment of GBM patients.

## 2. Current Approaches for the Diagnosis of GBM

Initial symptoms of GBM are often nonspecific [47], such as headaches, personality changes, and nausea. These may also resemble stroke symptoms. Symptoms can quickly worsen, potentially leading to the loss of consciousness and difficulty with swallowing, often arising in the week before death. Other common symptoms include progressive neurological deficits, incontinence, progressive cognitive deficits, and headache [48].

The initial diagnosis of GBM typically involves neuroimaging such as MRI or CT scans, followed by either the surgical resection or biopsy of tumor tissue to confirm the diagnosis, determine its grade, and characterize its pathological, genetic, genomic, transcriptomic, proteomic, and other molecular properties. Further tests are conducted on tumor samples using various pathological diagnosis methods such as immunohistochemistry (IHC) and molecular profiling methods such as transcriptomics, proteomics, and genomics [37,49,50], including assessments for the combined loss of chromosome arms 1p and 19q, mutations and/or the expression of p53, the presence of isocitrate dehydrogenase 1 (*IDH*1) mutation (commonly within exon 4 to codon 132, with the most frequent being c.395 G>A (R132H) substitutions) [51], and epigenetic modifications such as *MGMT* hypermethylation [29].

Tissue biopsies represent the gold standard method for diagnosing GBM; however, these procedures come with inherent risks to patients, including the potential for brain swelling in and around the tumor mass and the possibility of impacting neurological functions [43]. Moreover, the serial collection of tissue biopsies for the real-time and dynamic monitoring of tumor characteristics and treatment responses, as well as for predicting disease prognosis in GBM patients, may not be feasible. Additionally, some tumors may be challenging to access due to their location [52]. Moreover, tissue biopsies may not always accurately capture the heterogeneity of the entire tumor mass and may not provide a real-time representation of tumor activity [42]. Therefore, additional tests such as liquid biopsy-based circulating biomarkers might offer a complementary approach to the standard methods for risk stratification, monitoring disease progress and therapy responses in GBM patients.

As highlighted in the literature [53,54] and illustrated in Figure 1, the 2021 WHO classification of tumors of the CNS introduces significant changes. These include limiting the diagnosis of GBM to tumors that are *IDH* wild type, reclassifying previously diagnosed *IDH*-mutated GBMs as astrocytomas, *IDH*-mutated, and grade 4, and requiring the presence of *IDH* mutations for the classification of tumors as astrocytomas or oligodendrogliomas [53].

The WHO CNS5 recommends using diagnostic strategies that incorporate traditional histology, along with tissue-based tests such as immunohistochemistry (IHC) and ultrastructural analyses, as well as emerging molecular features [14,55,56]. Integrating essential genes, pathways, and molecules highlighted by WHO CNS5, which play a role in GBM pathogenesis, can further improve the development of precise diagnostic methods. A comprehensive list of CNS tumor-specific molecular markers can be found in the literature [14,55].

## 3. Current Standard of Care for Treating GBM

The current SoC for newly diagnosed GBM includes surgical resection, followed by radiotherapy (RT) and chemotherapy with temozolomide (TMZ). Adjuvant: A 4-week rest period after concurrent therapy. The dose could be reduced based on the appearance of toxicity [30,57]. For TMZ to be effective, the O6-Methylguanine-DNA methyltransferase (*MGMT*) promoter must be hypermethylated, which inhibits the expression of the *MGMT* gene responsible for repairing DNA damage [31]. This modification sensitizes tumor cells to TMZ’s DNA-damaging effects, enhancing its therapeutic efficacy [31]. Testing for *MGMT* promoter methylation status is crucial for predicting the response to TMZ therapy, and *MGMT* promoter methylation should be assessed as a continuous variable [58].

As for recurrent GBM, anti-VEGF therapy bevacizumab has been used. Treatment continues until disease progression or unacceptable toxicity.

Additional chemotherapy options for patients with newly diagnosed or recurrent GBM include nitrosourea drugs that alkylate DNA and RNA. These nitrosoureas include lomustine (CCNU), which is taken orally and used for adult GBM at disease recurrence, and for adult-type pediatric high-grade gliomas (HGG) at diagnosis when combined with TMZ. Another option is carmustine (BCNU), which is delivered using drug-impregnated wafers that are placed at the time of initial surgery or reoperation in the tumor cavity. It is worth noting that academic neurosurgeons have not favored the use of carmustine wafers [59,60].

The complete removal of all tumor cells during surgery is challenging due to the highly invasive nature of GBM cells in surrounding normal tissue. Consequently, GBM tumors often recur in most cases, with a median overall survival for patients with recurrent GBM of around 6.2 months [61]. Therapy resistance and high relapse rates contribute to this limited survival [30,32,33,34,35]. The emergence of resistance is primarily caused by tumor cells evading resection and/or invading normal brain parenchyma. GBM cancer stem cells (CSCs), a subset of tumor cells resistant to radiation and chemotherapy, may also drive rapid tumor recurrence [32,33,34,35]. Other factors, such as intra- and inter-tumor heterogeneity at the cellular and molecular levels, tumor plasticity, an inherently immunosuppressive TME, and tumor genomic characteristics, may also contribute to rapid tumor recurrence and relapse [32,33,34,35,62,63,64]. In addition, challenges involving the persistence and delivery of therapeutic antibodies and vaccines and the efficiency of drug penetration through the blood–brain barrier (BBB) continue to present significant challenges that need to be addressed.

To overcome these challenges, various novel immunotherapies, such as ICI nivolumab and VEGF inhibitor bevacizumab, are under investigation [36,37,38], although data on their efficacy in GBM treatment are mixed. Although the low tumor mutational burden (TMB) and immunologically cold nature of GBM pose challenges for IT [39], combining traditional ICIs may offer benefits by altering the TME [36,40,41]. For example, anti-PD-1 and anti-CTLA-4 therapies together show promising efficacy in treating recurrent GBM [41]. Traditional anti-PD-1 therapy may also be combined with other targets, such as TIM-3 and BTLA. Combining IT with another targetable mechanism is also being explored. For example, recent studies on combining bevacizumab with anti-CD276 have shown promise due to the established link between angiogenesis and CD276/B7-H3 [41].

Other innovative therapies, such as tumor-treating fields (TTFields), have been explored and have shown modest improvements in median survival for GBM patients [26].

Other novel treatments, such as tumor vaccines, including peptide-, mRNA-, and cell-based vaccines (e.g., dendritic cell vaccines and tumor cell vaccines), have been investigated, with some promising results [65,66,67,68,69,70]. Targeting neoantigens alone is challenging due to the low mutation burden in GBM, and single-peptide therapeutic vaccines have shown limited efficacy as standalone treatments. Thus, combining a variety of antigens as a vaccine cocktail such as neoantigens, tumor-associated antigens (TAAs), and pathogen-derived antigens along with optimizing vaccine design and vaccination strategies may enhance clinical efficacy [67]. Recent studies have demonstrated the potential utility of personalized cancer peptide vaccines targeting novel antigens [65,67,69]. In the first study, a personalized cancer vaccine targeting a novel antigen was created, which was identified by comparing the whole exon sequence data from the resected tumor with those of the matched normal tissues [65]. For each patient, 7 to 20 antigens that were predicted to have a high affinity for HLA type-I binding were chosen for vaccine development. In another study, two novel antigens and non-mutated tumor-associated antigens were combined to increase the number of binding epitopes [69]. Nine non-mutated peptides (APVAC1 patient) were included in a vaccine composition after injection, followed by the administration of 20 peptides of new antigens (APVAC 2). Both studies were phase I clinical trials; they could induce a considerable number of invasive tumor-reactive T memory cells and the clonal expansion of antigen-specific cells.

Another recent study has reported that mRNA vaccine therapy has shown promising safety and efficacy in preclinical studies involving mouse models and dogs with naturally occurring brain tumors, as well as in four adult GBM patients [68]. This mRNA vaccine approach triggered robust immune responses within 24–48 h, including rapid cytokine/chemokine release, immune activation/trafficking, tissue-confirmed pseudoprogression, and glioma-specific immune responses. The therapy works by rapidly reprogramming the TME, enabling simultaneously activated T cells to exert their effector functions after delivering mRNA vaccines encapsulated in multi-lamellar RNA lipid particle aggregates (LPAs) intravenously. Compared to the historical median progression-free survival (PFS) of 6 months [30], patients A25 and E42 had a progression-free survival of 8 months and 9 months, respectively. The 10 dogs had a median survival of 139 days, significantly longer than the typical 30 to 60 days for dogs with brain tumors. Once an optimal and safe dose is determined in a Phase I trial with 24 adult and pediatric patients, Phase II trials with approximately 25 children are planned to further validate these findings [68].

## 4. Current Approaches for the Prognosis of GBM

To assess the prognosis of GBM, brain MRI scans are conducted post-treatment, where contrast-enhancing lesions can indicate either tumor progression or pseudoprogression, the latter being post-radiotherapy changes that may resolve spontaneously [71]. Pseudoprogression affects 10–30% of GBM patients after their initial MRI scan, typically within 12 weeks of treatment [71]. Differentiating between true progression and pseudoprogression is essential because it can help avoid unnecessary surgeries and ineffective treatments [43,46,49,71,72]. Currently, there are no validated biomarkers or clinical features for distinguishing true progression from pseudoprogression. A recent study [73] has demonstrated that patients with methylation of the *MGMT* gene promoter exhibited higher rates of pseudoprogression (91%) compared to those with unmethylated *MGMT* (41%). Similarly, p53 overexpression in tumor tissue was correlated with pseudoprogression in glioma patients [74]. Further research suggested that elevated expressions of X-ray repair cross-complementing 1 (*XRCC1*) and interferon regulatory factor 9 (*IRF9*) were associated with pseudoprogression [75]. Despite these findings, additional studies are needed to identify minimally invasive and reliable circulating biomarkers for clinical use in distinguishing true progression from pseudoprogression.

## 5. Liquid Biopsies in Cancer

As highlighted in recent literature [76], considering the limitations of MRI and tissue biopsies outlined earlier, there is an urgent and unmet clinical need for identifying and validating alternative and complementary techniques aiding in the diagnosis, risk stratification, and real-time and dynamic monitoring of tumor characteristics, treatment responses, and disease prognosis through the repeated sampling of GBM patients.

As highlighted in the recent literature [77,78] and illustrated in Figure 2, in the subsequent section, we will delve into the biological foundations, benefits, and drawbacks of various circulating biomarkers proposed for GBM.

Recent literature [56,76,78] highlights the application of liquid biopsy, mainly via blood tests, which includes detecting and quantifying the tumoral content released by tumors into biofluids such as blood, CSF, saliva, vitreous, and urine [56,79]. Tumors release their content into various body fluids such as the bloodstream, CSF, and other biofluids which can be frequently sampled for the real-time analysis of circulating biomarkers [80] including ctDNA, CTCs, miRNAs, EVs, proteins, metabolites, and others. The process of sampling and analyzing these molecules in non-solid biological fluids is known as a liquid biopsy [81] or fluid-phase biopsy [82].

Although blood draws are common for liquid biopsies, other fluids such as CSF, saliva, urine, and cyst fluid can also be used [79]. For example, a recent study has demonstrated that cfDNA from cyst fluid in cystic brain tumors is a reliable alternative to tumor DNA for diagnosing brain tumors [83]. CSF has been utilized to study tumor-specific biomarkers in brain tumors [44,84] due to its proximity to the CNS. In pediatric patients with tumors, particularly medulloblastoma and other embryonal tumors, CSF sampling via post-operative lumbar puncture is a standard part of staging. This procedure is considered minimally invasive, often performed under conscious sedation or general anesthesia. Recent evidence suggests that for diffuse midline gliomas (DMG H3K27-altered), especially those in the pons, CSF-derived cfDNA serves as a surrogate biomarker for measurable residual disease (MRD) [85]. Serial CSF samples collected from children with medulloblastoma are more reliable for analysis than blood, serum, or plasma [85]. However, CSF collection involves a minimally invasive procedure. In contrast, blood-based liquid biopsies offer a less invasive method for the serial sampling of blood samples to monitor tumor activity in real time for predicting the therapy response and disease progression [79,86]. As a result, liquid biopsies for exploring circulating factors have been explored in various cancer types [87] such as breast [88], head and neck [89], lung [90], and pancreatic cancers [77], among others. In lung cancer, for instance, blood plasma can detect mutations in the *EGFR* gene when the tumor tissue is limited [91,92]. The FDA has approved a pan-cancer diagnostic test using ctDNA from liquid biopsies to detect multiple solid tumors (e.g., non-small-cell lung cancer, colorectal cancer, breast cancer, ovarian cancer, and melanoma) [93,94].

In the context of GBM, the successful use of liquid biopsies relies on tumor-specific material crossing the blood–brain barrier (BBB), which regulates the exchange of nutrients, vitamins, and other molecules in the brain [95]. BBB dysfunction plays a significant role in the pathogenesis of various brain disorders. The integrity of the BBB is crucial for a healthy brain environment, with disruptions linked to GBM progression. Hypoxia in GBM contributes to BBB disruption, allowing tumor-specific material to cross the BBB. Various signaling factors, such as inflammatory mediators, free radicals, vascular endothelial growth factor, matrix metalloproteinases, and miRNAs, regulate BBB permeability by affecting structural components like tight junction proteins, integrins, annexins, and agrin within a complex multicellular environment or system that includes endothelial cells, astrocytes, pericytes, etc. [95].

Studies have shown that EVs derived from GBM cells can cross the intact BBB [96], facilitating the passage of biomarkers into the bloodstream, even when the BBB is intact. As a result, liquid biopsies provide the real-time and dynamic monitoring of tumor characteristics and treatment responses, enabling the prediction of GBM prognosis and the assessment of chemotherapy effectiveness through repeated sampling [79,86,97].

Liquid biopsies can detect and quantify various types of biomarkers: CTCs released from a primary tumor; EVs, which may carry nucleic acids and proteins and can be released by tumor cells; as well as ctDNA and miRNAs, which can also be released by tumor cells. These molecules carry tumoral information (e.g., mutational status, tumoral cargo), which can be sampled non-invasively.

As highlighted in the recent literature [78] and illustrated in Figure 2, which depicts a schematic representation of biomolecular transportation from a tumor through the BBB into the circulation, various biomarkers can be detected and measured in circulation. Additionally, Figure 2 outlines the advantages and disadvantages of liquid biopsy compared to tumor tissue or biopsy.

Analytes like nucleic acids (ctDNAs/mRNAs, non-coding RNAs such as miRNAs), proteins, and metabolites can be obtained from circulating cell-free sources or extracted from CTCs, EVs, and tumor-educated platelets [56]. Each of these circulating analytes presents opportunities for investigating tumor-specific changes, including various types of mutations, epigenetic alterations, DNA fragmentation patterns, nucleosome organization, chromosomal abnormalities, changes in the levels of RNAs/proteins/metabolites, and post-translational modifications [56].

Table 1 illustrates examples of several prospective clinical studies that are currently exploring the potential of liquid biopsies as diagnostic, predictive, and prognostic biomarkers in GBM.

However, there are differences in the detection sensitivity, specificity, and reproducibility of each class of circulating biomarkers.

In addition, while most of these biomarkers have a short half-life and degrade quickly in plasma [46,98], some are protected within EVs like microvesicles and exosomes, shielding them from degradation [46].

Recent literature [77] and Figure 3 summarize the comparison of liquid biopsy techniques, including their capabilities, shortcomings, and available technologies.

## 6. ctDNA Profiling as a Potential Biomarker for GBM

The first time the existence of cell-free nucleic acids, including cell-free DNA (cfDNA), in the blood of healthy individuals and patients with different metabolic or oncological disorders was reported was in 1948 by Mandel and Metais [99]. Subsequently, elevated levels of cfDNA in the serum of patients with cancer compared to those of healthy individuals were first reported in 1977 by Leon et al. [100]. Similarly, Stroun et al. [101] reported neoplastic characteristics (i.e., decreased strand stability of cancer cell DNA) found in the cfDNA of cancer patients. Subsequent research validated the presence of various tumor-related genomic aberrations, including mutations in oncogenes and tumor-suppressor genes [100], epigenetic modifications [102], and microsatellite instability statuses [103]. cfDNA, released into the circulation by tumor cells carrying the genetic and epigenetic alterations of the original tumor, is termed circulating tumor DNA (ctDNA) [104].

Despite the demonstration that ctDNA exhibits high specificity to the tumor from which it was derived, reflected by a strong agreement between the mutational profile of ctDNA and matched tumor tissue across various cancers [105,106,107], the mechanisms underlying the release of circulating ctDNA into the bloodstream remain unclear. One proposed mechanism for the source of ctDNA is the apoptosis of neoplastic cells, triggered by factors such as hypoxia, which generates DNA fragments typically ranging from 130 to 180 base pairs. This process involves the activity of a caspase-activated DNase that degrades chromatin into mono- and oligonucleosomes [108,109]. The necrosis of tumor cells is another proposed mechanism for the release of ctDNA into bodily fluids. ctDNA resulting from necrosis is generally larger in size compared to that originating from apoptosis [109]. For example, a recent study [110] demonstrated that tumor size and cell proliferation impact ctDNA release in patient-derived orthotopic xenograft mice models before treatment, with no significant influence from BBB integrity. However, they noted that post-therapy, cell death contributes to increased ctDNA release. These findings challenge the notion that BBB integrity predominantly regulates ctDNA release, as suggested in earlier studies. Additionally, macrophages can release DNA fragments following the engulfment of necrotic cancer cells [111]. Fragmented DNA released by healthy cells (i.e., cfDNA) is cleared through phagocytosis, resulting in a generally low background level of cfDNA in circulation, with an average concentration of 30 ng/mL [42,112].

In cancer patients, the mechanisms responsible for clearing DNA fragments are overwhelmed by those released from tumor cells. Consequently, a proportion of circulating cfDNA, ranging from as little as 0.01% to as high as 90%, consists of ctDNA [42]. Notably, the background level of cfDNA is higher in serum than in plasma, likely due to contamination with DNA released by immune cells during the clotting process. Therefore, plasma samples are preferred for ctDNA studies [113].

Two primary methods are used for detecting mutations in ctDNA: polymerase chain reaction (PCR)-based techniques targeting known point mutations and next-generation sequencing (NGS) or whole genome sequencing (WGS) techniques enabling the detection of novel and unknown mutations [72]. In addition, a recent study has demonstrated that copy number analysis can be effectively performed with the multiplex ligation-dependent probe amplification of cfDNA in CSF from patients with adult diffuse glioma [114].

As highlighted in the recent literature [76], examples of several studies investigating ctDNA in GBM are shown in Table 2. In these studies, the number of patients in each study was relatively small, particularly when CSF was used due to the relatively invasive nature of its collection. However, it has been reported that the detection rate of ctDNA is higher in CSF compared to that in plasma and serum. This could be attributed to the partial disruption of the BBB, which still restricts the passage of primary tumor-derived ctDNA into the bloodstream [115]. Other factors may include the shorter distance for ctDNA to travel before sampling, less efficient ctDNA clearance mechanisms, and the lower background cfDNA levels in CSF compared to those in blood [78,116].

Despite the encouraging findings, utilizing ctDNA as a biomarker, especially for GBM, presents challenges. (1) The quantity of ctDNA varies depending on the tissue type and cancer stage, with higher levels typically seen in advanced-stage cancers, limiting its potential primarily to early-stage diagnosis [98]. (2) Gliomas exhibit among the lowest detectable levels of ctDNA [98]. (3) ctDNA has a short half-life (<2.5 h), necessitating rapid processing post-sampling [117]. (4) Even when detectable, its concentration in cancer is very low (180 ng/mL) and potentially even lower in GBM cases, demanding highly sensitive techniques for its accurate identification and differentiation from normal tissue cfDNA [112]. Despite these challenges, several prospective clinical studies are currently investigating the potential of circulating ctDNA as a diagnostic, predictive, and prognostic biomarker in GBM (Table 3).

**Table 2 ijms-25-07974-t002:** Examples of studies reporting cfDNA and ctDNA in CNS tumors including GBM. Only studies in which data for GBM patients were available are reported.

Biomarker	Study Title	Cancer Types	Patients (*n*)	Control	Biofluid	Method	ctDNA Detection Rate	Gene Panel	Alterations	Results	References
ctDNA	Detection rate of actionable mutations in diverse cancers using a biopsy-free (blood) circulating tumor cell DNA assay	Lung (23%), breast (23%), glioblastoma (19%).	171	222 healthy volunteers	Plasma	NGS	27%	Guardant 360 54 genes and CNVs in *EGFR, ERBB2*, and *MET*	*TP53* (29.8%), *EGFR* (17.5%), *MET* (10.5%), *PIK3CA* (7%), and *NOTCH1* (5.8%)	69 patients had actionable alterations (40% of the total; 69.7% of patients (69/99) with alterations); 68 patients (40% of the total; 69% of patients with alterations)	[118]
cfDNA	Analysis of cell-free circulating tumor DNA in 419 patients with glioblastoma and other primary brain tumors	Glioblastoma, meningioma	419	NA	Plasma	NGS	55%	Guardant 360	SNVs in 61 genes, with amplifications in *ERBB2, MET, EGFR*, and others	Detection was highest in meningioma (59%) and glioblastoma (55%). SNVs were detected in 61 genes, with amplifications detected in ERBB2, MET, EGFR, and others	[119]
cfDNA	The Landscape of Actionable Genomic Alterations in Cell-Free Circulating Tumor DNA from 21,807 Advanced Cancer Patients	Late-stage cancers across >50 cancer types	Total: 21,807 GBM: 107		Plasma	NGS	51%	Guardant 360	*EGFR* and *ERBB2*	cfDNA clonality and copy-number driver identification methods revealed significant mutual exclusivity among predicted truncal driver cfDNA alterations for EGFR and ERBB2, in effect distinguishing tumor-initiating alterations from secondary alterations. Dataset reveals subclonal structures and emerging resistance in advanced solid tumors	[120]
cfDNA	Clinical Utility of Plasma Cell-Free DNA in Adult Patients with Newly Diagnosed Glioblastoma: A Pilot Prospective Study	Newly diagnosed GBM	42		Plasma	NGS	55%	152-gene panel (Comprehensive Solid Tumor HaloPlexHS, version 2.0; Agilent Technology, Inc., Santa Clara, CA, USA Guardant 360)	Plasma cfDNA concentration was correlated with radiographic tumor burden. Preoperative plasma cfDNA concentration above the mean (>13.4 ng/mL) was associated with inferior PFS (median 4.9 vs. 9.5 months, *p* = 0.038). Detection of ≥1 somatic mutation in plasma cfDNA occurred in 55% of patients and was associated with nonstatistically significant decreases in PFS (median 6.0 vs. 8.7 months, *p* = 0.093) and OS (median 5.5 vs. 9.2 months, *p* = 0.053)		[121]
ctDNA	Plasma cell-free circulating tumor DNA (ctDNA) detection in longitudinally followed glioblastoma patients using *TERT* promoter mutation-specific droplet digital PCR assays		13		Plasma	ddPCR	46%	*TERT* promoter mutations (7 C228T and 6 C250T		13/14 (92.9%) *IDH*wt tumors had *TERT* mutations (7 C228T and 6 C250T). Six of these thirteen (46%) pts had positive plasma *TERT* ctDNA preop (4 C228T, 2 C250T). Detected plasma *TERT* ctDNA in 46% of *TERT* mutant GBM pts before surgery and in 100% of pts with multiple contrast-enhancing lesions. *TERT* mutant ctDNA levels correlated with pseudoprogression or true disease progression and predicted progression before MRI	[122]
ctDNA	*MGMT* promoter methylation in serum and cerebrospinal fluid as a tumor-specific biomarker of glioma	32 WHO grade II, 19 WHO grade III, and 38 WHO grade IV were pathologically diagnosed as glioma	89		Serum, CSF, tissue	Methylation-specific PCR assay	37% (Serum), 61% (CSF)	*MGMT* promoter methylation		Among the tumor tissue samples, 51/89 (57.3%) showed *MGMT* promoter methylation. The specificity of the detection in the CSF and serum samples reached 100%. The sensitivity of *MGMT* promoter methylation detection in CSF and serum was 26/40 (65.0%) and 19/51 (37.3%), respectively (*p* < 0.05).	[123]
ctDNA	*TERT* Promoter Mutation Detection in Cell-Free Tumor-Derived DNA in Patients with *IDH* Wild-Type Glioblastomas: A Pilot Prospective Study	Glial tumors	Glial tumors: 60 Glioblastoma: 38		Plasma, CSF	Nested PCR	8% (Plasma), 92% (CSF)	*TERT* promoter (*TERT*p)-mutation	High *TERT*p mutation VAF levels in the CSF-tDNA could be a predictor of poor survival in GBM patients	The matched *TERT*p mutation in the CSF-tDNA was detected with 100% specificity (95% CI, 87.6–100%) and 92.1% sensitivity (95% CI, 78.6–98.3%) (*n* = 35/38). The sensitivity in the plasma-tDNA was lower [*n* = 3/38, 7.9% (95% CI, 1.6–21.4%)]. Observed a longer OS of patients with low VAF in the CSF-tDNA compared with patients with high VAF, irrespective of using the lower-quartile VAF.	[124]
cfDNA	Detection of tumor-derived DNA in cerebrospinal fluid of patients with primary tumors of the brain and spinal cord		35 primary CNS malignancies including medulloblastomas, ependymomas, and high-grade gliomas (*n* = 11)		CSF	WGS	100%		Detected at least one mutation in each tumor using targeted or genome-wide sequencing	Detected cfDNA in 74% of cases. All primary CNS tumors that were directly adjacent to a CSF space were detectable (100% of 21 cases; 95% CI = 88–100%), whereas no cfDNA was detected in patients whose tumors were not directly adjacent to a CSF reservoir (*p* < 0.0001, Fisher’s exact test)	[125]
cfDNA	Detection of cfDNA fragmentation and copy number alterations in CSF from glioma patients		13	NA	CSF	WGS	50%		Detection of somatic copy number alterations and DNA fragmentation patterns	Detected the presence of cfDNA in CSF without any prior knowledge of point mutations present in the tumor. Identified somatic copy number alterations in 5/13 patients. The fragmentation pattern of cfDNA in CSF is different from that in plasma.	[126]
ctDNA	Molecular Diagnosis of Diffuse Gliomas through Sequencing of Cell-Free Circulating Tumor DNA from Cerebrospinal Fluid	Gliomas	The TCGA cohort including 648 diffuse gliomas. CSF and tumor samples from 20 diffuse glioma patients		CSF and tumor	ddPCR	100%		Analysis of *IDH1*, *IDH2*, *TP53*, *TERT*, *ATRX*, *H3F3A*, and *HIST1H3B* gene mutations	The mutational status of the *IDH1*, *IDH2*, *TP53*, *TERT*, *ATRX*, *H3F3A*, and *HIST1H3B* genes allowed for the classification of 79% of the 648 diffuse gliomas analyzed into *IDH*-wild-type glioblastoma, *IDH*-mutant glioblastoma/diffuse astrocytoma, and oligodendroglioma, each subtype exhibiting diverse median overall survival (1.1, 6.7, and 11.2 years, respectively).	[127]
ctDNA	Tracking tumour evolution in glioma through liquid biopsies of cerebrospinal fluid	Glioma	85		CSF	NGS	59%		Chromosome arms 1p and 19q (1p/19q codeletion) and mutations in *IDH1* or *IDH21,2* growth factor receptor signaling pathways	Tumor-derived ctDNA was detected in CSF from 42 out of 85 patients (49.4%) and was associated with a disease burden and adverse outcome. The genomic landscape of glioma in the CSF revealed various genetic alterations and resembled the genomes of tumor biopsies. Co-deletion of chromosome arms 1p and 19q (1p/19q codeletion) and mutations in *IDH1* or *IDH21,2* were shared in all matched ctDNA-positive CSF-tumor pairs. Contrastingly, growth factor receptor signaling pathways showed considerable evolution	[44]
cfDNA	Cerebrospinal fluid cfDNA sequencing for classification of central nervous system glioma	Primary or recurrent glioma	85 CSF and matching 38 tumor samples						CNVs, SNVs, and Indels	Cancer-specific alterations in 75% (*n* = 24) of GBM and 52.6% (*n* = 10) of other glioma cases. The overlap between CSF and matching solid tumor tissue was highest for CNVs (26–48%) and SNVs at pre-defined gene loci (44%), followed by SNVs/indels identified via uninformed variant calling (8–14%)	[128]
cfDNA	Analysis of cell-free circulating tumor DNA in 419 patients with glioblastoma and other primary brain tumors	Primary brain tumors including GBM	419 primary tumors including 222 GBM							Detected ctDNA mutations in blood samples collected from 50% of all brain-tumor patients—55% among the GBM patients	[119]
ctDNA	Detection of EGFRvIII mutant DNA in the peripheral blood of brain tumor patients	Newly diagnosed with GBM	13		Plasma			EGFRvIII mutation		ctDNA status for EGFRvIII correlates with the analysis of the tumor samples, and its level correlates with the extent of the tumor resection	[129]
cfDNA	Circulating cell-free D. N. A. as a prognostic and molecular marker for patients with brain tumors under perillyl alcohol-based therapy.	Patients at terminal stages with GBM, *n* = 122 Brain metastasis from stage IV adenocarcinomas, *n* = 55	(GBM, *n* = 122) or brain metastasis (*n* = 55) from stage IV adenocarcinomasControls: 130 healthy subjects		Serum			Serum cfDNA levels		Compared to controls (40 ng/mL), patients with brain tumors before ITN-POH treatment had increased (*p* < 0.0001) cfDNA median levels: GBM (286 ng/mL) and brain metastasis (588 ng/mL). ITN-POH treatment was significantly correlated with a survival of >6 months at a concentration of 599 ± 221 ng/mL and of <6 months at 1626 ± 505 ng/mL, but a sharp and abrupt increase in cfDNA and tumor recurrence occurred after ITN-POH discontinuation. Patients undergoing ITN-POH treatment and checked with brain MRI compatible with CR had cfDNA levels similar to those of the controls	[130]

Abbreviations: cfDNA, cell-free DNA; ctDNA, circulating tumor DNA; CNVs, copy number variants; CR, complete response; CSF, cerebrospinal fluid; ddPCR, (droplet digital) polymerase chain reaction; GBM, glioblastoma; Indels, insertions/deletions; ITN-POH, intranasal administration (ITN) of perillyl alcohol (POH); MRI, magnetic resonance imaging; NGS, next-generation sequencing; OS, overall survival; SNVs, single nucleotide variants; TCGA, the Cancer Genome Atlas; VAF, variant allele frequency; WGS, whole genome sequencing; WHO, World Health Organization.

**Table 3 ijms-25-07974-t003:** Recent clinical studies that evaluated ctDNA as a potential biomarker in GBM. A search was conducted on Clinicaltrials.gov using the terms “ctDNA”, “circulating tumor DNA”, and “Glioblastoma” on 6 March 2024.

Rank	NCT Number	Title	Status	Study Results	Conditions	Interventions	Phases	Study Type	URL
1	NCT05539339	Personalized Trial in ctDNA-level-relapse Glioblastoma	Not yet recruiting	No Results Available	Glioblastoma	Other: Individualized intervention based on genomic alterations	Not Applicable	Interventional	https://ClinicalTrials.gov/show/NCT05539339
2	NCT03115138	Evaluation of Circulating Tumor DNA as a Theranostic Marker in the Management of Glioblastomas.	Terminated	No Results Available	Glioblastoma, Molecular Disease	Other: Correlation between molecular anomalies of the primary tumor and circulating tumor DNA	Not Applicable	Interventional	https://ClinicalTrials.gov/show/NCT03115138
3	NCT05502991	Sintilimab (One Anti-PD-1 Antibody) Plus Low-dose Bevacizumab for ctDNA-level-relapse and Clinical-relapse Glioblastoma	Not yet recruiting	No Results Available	Glioblastoma	Drug: Tislelizumab plus Bevacizumab	Phase 2	Interventional	https://ClinicalTrials.gov/show/NCT05502991
4	NCT05541042	Radiogenomics in Glioblastoma: Correlation Between Multiparametric Imaging Biomarkers and Genetic Biomarkers	Not yet recruiting	No Results Available	Glioblastoma	Other: Observational only		Observational	https://ClinicalTrials.gov/show/NCT05541042
5	NCT05695976	GRETeL: Tumor Response to Standard Radiotherapy and TMZ Patients With GBM	Recruiting	No Results Available	Glioblastoma, Glioma, Malignant			Observational	https://ClinicalTrials.gov/show/NCT05695976
6	NCT05281731	Sonobiopsy for Noninvasive and Sensitive Detection of Glioblastoma	Recruiting	No Results Available	Glioblastoma, Glioblastoma Multiforme	Device: Sonobiopsy, Procedure: Research blood, Genetic: Cancer Personalized Profiling, Device: Definity^®^	Not Applicable	Interventional	https://ClinicalTrials.gov/show/NCT05281731
7	NCT04776980	Multimodality MRI and Liquid Biopsy in GBM	Withdrawn	No Results Available	Glioblastoma Multiforme, Brain Tumor, Adult: Glioblastoma, Brain Tumor, Recurrent, Brain Tumor, Primary	Diagnostic Test: Post-Feraheme Infusion MRI	Early Phase 1	Interventional	https://ClinicalTrials.gov/show/NCT04776980
8	NCT04868396	Patient-derived Glioma Stem Cell Organoids	Active, not recruiting	No Results Available	Glioblastoma	Procedure: Tumor biopsy		Observational	https://ClinicalTrials.gov/show/NCT04868396
9	NCT05540275	Tislelizumab (One Anti-PD-1 Antibody) Plus Low-dose Bevacizumab for Bevacizumab Refractory Recurrent Glioblastoma	Not yet recruiting	No Results Available	Recurrent Glioblastoma	Drug: Tislelizumab plus Bevacizumab	Phase 2	Interventional	https://ClinicalTrials.gov/show/NCT05540275
10	NCT05099068	Profiling Program of Cancer Patients with Sequential Tumor and Liquid Biopsies (PLANET)	Recruiting	No Results Available	Advanced/Metastic Solid Tumors, Glioblastoma, Chronic Leukemia Lymphocytic	Biological: Blood and tumor samples	Not Applicable	Interventional	https://ClinicalTrials.gov/show/NCT05099068
11	NCT02060890	Molecular Profiling in Guiding Individualized Treatment Plan in Adults with Recurrent/Progressive Glioblastoma	Completed	Has Results	Adult Glioblastoma	Other: specialized tumor board recommendation		Observational	https://ClinicalTrials.gov/show/NCT02060890
12	NCT05934630	Testing Cerebrospinal Fluid for Cell-free Tumor DNA in Children, Adolescents, and Young Adults with Brain Tumors	Active, not recruiting	No Results Available	Anaplastic Astrocytoma, Diffuse Brainstem Glioma, Glioblastoma Multiforme, High-grade Astrocytoma NOS, Fibrillary Astrocytoma, Low-Grade Astrocytoma, Nos, Pilocytic Astrocytoma, Choroid Plexus Carcinoma, CNS Primary Tumor, Nos, Atypical Teratoid/Rhabdoid Tumor, Medulloblastoma, Supratentorial Primitive Neuroectodermal Tumor, Ependymoma, NOS, Anaplastic Oligodendroglioma, Oligodendroglioma, Nos, CNS Germ Cell Tumor, Pineoblastoma, Diffuse Leptomeningeal Glioneuronal Tumor			Observational	https://ClinicalTrials.gov/show/NCT05934630
13	NCT03973918	Study of Binimetinib with Encorafenib in Adults with Recurrent *BRAF V600*-Mutated HGG	Terminated	Has Results	High-Grade Glioma, *BRAF V600E*, *BRAF V600K*, Anaplastic Astrocytoma, Anaplastic Pleomorphic Xanthoastrocytoma, Gliosarcoma, Glioblastoma	Drug: Encorafenib, Drug: Binimetinib, Biological: Research Bloods, Biological: Tumor Tissue	Phase 2	Interventional	https://ClinicalTrials.gov/show/NCT03973918
14	NCT04888611	Neoadjuvant PD-1 Antibody Alone or Combined with DC Vaccines for Recurrent Glioblastoma	Recruiting	No Results Available	Recurrent Glioblastoma	Biological: Camrelizumab plus GSC-DCV, Biological: Camrelizumab plus Placebo	Phase 2	Interventional	https://ClinicalTrials.gov/show/NCT04888611
15	NCT04528680	Ultrasound-based Blood-brain Barrier Opening and Albumin-bound Paclitaxel and Carboplatin for Recurrent Glioblastoma	Recruiting	No Results Available	Glioblastoma, Gliosarcoma, GBM, Glioblastoma Multiforme, Glioblastoma, *IDH*-wildtype, Recurrent Glioblastoma	Device: Sonication for opening of the blood–brain barrier, Drug: Chemotherapy, albumin-bound paclitaxel, Drug: Chemotherapy, carboplatin	Phase 1, Phase 2	Interventional	https://ClinicalTrials.gov/show/NCT04528680

## 7. Circulating miRNA Profiling as a Potential Biomarker

Research indicates that miRNAs regulate gene expression at both transcriptional and post-transcriptional levels, playing crucial roles in a wide range of biological processes within cells and organisms. Consequently, dysregulated miRNA expression has been linked to various pathological processes and the development of diseases such as diabetes, cardiovascular diseases, neurodegenerative diseases, cancer, and other malignancies [56,77,78,87,131,132,133,134,135,136,137,138,139,140,141,142,143,144,145,146,147,148,149,150,151].

The discovery that many miRNAs can be detected in cell-free conditions in biofluids like blood (serum or plasma), CSF, saliva, urine, etc. and that they exhibit specific expression patterns associated with different physiological and disease states [132,137,138,139,140,141,152,153,154] renders them promising candidates as biomarkers for diagnosing, prognosing, and monitoring the treatment of various human malignancies.

Further studies have shown that miRNAs can be transported to other cells via EVs such as exosomes, microvesicles, and apoptotic bodies under various physiological and pathological conditions, acting as chemical messengers for cell-to-cell communication [153,155,156], or by binding to proteins like Argonautes, specifically AGO2 [155,157], thereby regulating gene transcription and translation [158], which assigns additional roles to miRNAs.

Data also indicate that significant portions of circulating miRNAs are contained within EVs, exosomes, and various cell types such as tumor cells [159], stem cells [160], macrophages [161], and adipocytes [162], all of which release exosomes with specific miRNA (exomiR) content into the circulation.

Moreover, dysregulated circulating miRNAs have been associated with the disease origin, progression, treatment response, and patient outcome and survival [163,164]. Thereby, the unique tissue specificity of miRNAs [165], essential for maintaining normal cell and tissue function [131], makes them potential biomarkers for diagnosing cancers of unknown primary origin [166,167].

As for CNS tumors, numerous studies have identified specific miRNAs with potential as diagnostic biomarkers for GBM [56,149,150,151]. For example, a recent study [146] highlighted that miR-21, miR-124-3p, and miR-222 collectively demonstrated a sensitivity of ~84% and specificity of ~86%. These microRNAs are particularly associated with advanced-stage GBM, effectively distinguishing disease progression from stable conditions. Moreover, these miRNAs displayed significant decreases in post-surgical resection in high-grade gliomas.

As highlighted in the literature [77,87,132,147,148], miRNAs were first identified in 1993 in *Caenorhabditis elegans* [168], constituting the most prevalent small RNAs, typically 21–23 nucleotides in length [169]. These single-stranded, non-coding RNAs regulate around 30% of protein-coding genes in the genome, predominantly by modulating gene expression post-transcriptionally through mRNA binding, resulting in translational inhibition or mRNA degradation [169]. Their involvement extends to various physiological and pathological processes, including cancer.

In GBM patients, miRNAs can be detected in circulation as cell-free nucleic acids in the blood and CSF sometimes encapsulated within extracellular vesicles (EVs), which enhances their stability [78]. As reported recently in the literature [148], miRNAs’ role in functional mechanisms and relevant signaling pathways in GBM has been studied in GBM tissues and cells. These studies have revealed that the upregulation of certain pro-oncogenic miRNAs promotes proliferation, cell cycle progression, aggressiveness, migration, and tumor cell differentiation. Conversely, inhibiting the expression of tumor-suppressive miRNAs promotes GBM progression, suppresses apoptosis, and correlates with a poor prognosis. Moreover, research has highlighted the involvement of miRNAs as regulators in various aspects of GBM biology, including communication within EVs, the modulation of immune responses, adaptation to the hypoxic microenvironment, and the response to reverse pH conditions [148].

As shown in Table 4 and highlighted in the recent literature [76,148,170], several clinical studies have identified dysregulated circulating miRNAs in patients with GBM, offering potential diagnostic and monitoring biomarkers for GBM. Examples of several clinical and preclinical studies investigating miRNAs in GBM are shown in Table 4 and Table 5 and highlighted in the literature [76,148,170], respectively. Besides distinguishing GBM patients from healthy individuals (Table 4), these studies also revealed that changes in the expression of specific miRNAs (upregulation: miR-210, miR-454-3p, miR-182, miR-20a-5p, miR-106a-5p, miR-181b-5p; downregulation: miR-128, miR-342-3p, miR-16, miR-497, miR-125b, miR-205) could effectively differentiate between patients with GBM and those with lower-grade gliomas or other brain pathologies [171,172,173,174,175,176,177,178,179,180], with reported sensitivities and specificities ranging from 58% to 99% and from 67% to 100%, respectively [171,172,173,174,175,176,177,178,179,181,182].

Variations in miRNA expression levels during the disease course, including before and after treatment or at recurrence, provide valuable insights for GBM management. Recent studies have highlighted several miRNAs as potential biomarkers for GBM, notably miR-17-3p, miR-222, and miR-340, identified through integrative analyses of large-scale genomic databases such as the TCGA and the GBM transcriptomes from biopsies [217].

Similarly, recent research [170] involving five studies have identified miRNAs that may contribute to the pathogenesis and progression of GBM through activities such as cell proliferation, invasion, and/or motility: miR-7 [183], miR-9 [183,218], miR-21 [183], miR-130b [189], miR-181c [219], miR-4725 [220], and miR-146b [221]. However, in recurrent GBM samples, a change in the expression pattern of miR-7 was observed, indicating heterogeneity among tumors regarding this molecule. Conversely, miR-9 did not show a change in the expression profile in recurrent samples. Their differential expression profiles in GBM tissues and recurrent tumors underscore their heterogeneity and potential clinical significance.

In a separate in vitro study, miR-9 expression was analyzed [218], revealing its involvement in cellular mobility and its potential role in controlling tumor progression by reducing migration and invasion activities linked to metastasis. These findings suggest that miR-9 may contribute to determining the progression of GBM. miR-21, known to influence apoptosis, invasion, proliferation, and chemoresistance pathways, exhibits decreased expression following treatment, possibly explaining GBM cell resistance to therapy. Similarly, miR-130b, upregulated in GBM tissues and cells, promotes tumor development and progression by enhancing cell migration, invasion, and proliferation [189]. However, the precise roles of miR-130b vary across different cancer types. Therefore, targeting Peroxisome Proliferator-activated Receptor Gamma (*PPAR-γ*) to silence miR-130b could potentially reverse its effects and serve as a therapeutic strategy for GBM. Likewise, miR-146b-5p has been associated with cell proliferation. Research indicates that elevated levels of miR-146b-5p decrease cellular activity and induce apoptosis [221] by inhibiting the *TRAF6-TRAK1* pathway. Conversely, reduced levels of the miR-146 family can contribute to tumorigenesis in GBM and other cancer types [221].

Moreover, studies elucidating the epigenetic regulation of miRNAs have identified associations between miR-181c and key regulatory factors such as the CCCTC-binding factor (CTCF) [219]. In GBM cell lines, miR-181c is downregulated compared to normal brain tissue. This downregulation correlates with increased DNA methylation at its promoter region and the loss of CTCF binding. The findings suggest CTCF may regulate miR-181c locally and in a cell type-specific manner, rather than through chromatin loop formation. This is supported by the depletion of CTCF in GBM cells, which influences the expression of *NOTCH2*, a target of miR-181c [219]. Therefore, the dysregulation of miRNA expression due to epigenetic alterations underscores the intricate regulatory networks governing tumor suppressor pathways in GBM.

Additionally, miR-4725 has been implicated in GBM progression by targeting stromal interacting molecule 1 (*STM1*) [220], an oncogene, highlighting the potential of miRNA-based therapies in disrupting tumor progression pathways.

Despite their potential, the clinical utility of miRNAs as circulating biomarkers is somewhat constrained by limitations such as small cohort sizes and the absence of standardized methodologies for blood collection, RNA extraction, and sequencing. Additionally, miRNAs generally exhibit lower specificity compared to ctDNA. Therefore, additional large-scale prospective studies are needed to validate the diagnostic potential of circulating miRNAs for GBM. Consequently, the specific miRNAs that can be used as biomarkers in the early screening stage of GBM are still being elucidated [133]. Table 6 summarizes examples of several recent prospective clinical studies that explored the potential of circulating miRNAs as diagnostic, predictive, and prognostic biomarkers in GBM. In summary, a better understanding of the mechanisms underlying miRNA dysregulation in GBM pathogenesis offers promising avenues for developing targeted therapies and overcoming therapeutic resistance. In conclusion, circulating miRNAs hold promise as diagnostic biomarkers for glioblastoma. Nonetheless, further research is necessary to confirm their efficacy and pinpoint specific miRNAs suitable for early screening purposes [133].

## 8. CTCs as Potential Biomarkers

Metastasis, the leading cause of cancer-related death, occurs when cancer cells dissociate from primary tumors, migrate to distant sites, and colonize, forming metastatic tumors. Circulating tumor cells (CTCs), such as those found in peripheral blood or bone marrow, play a role in this process [222].

Both the enumeration and molecular analysis of CTCs hold promise as methods for gaining insights into the biology of metastatic cancers, monitoring disease progression, assessing treatment responses, and guiding individualized treatment decisions [223]. However, their application in early cancer detection is more challenging compared to that of other liquid biopsy-based methods such as circulating tumor DNA (ctDNA). CTCs are rare and phenotypically heterogeneous and distinct subsets of the tumor cell population released by primary or metastatic lesions into biofluids [222,223]. In addition, CTCs exhibit heterogeneity at multiple levels, with only a small fraction capable of initiating metastasis [223].

Earlier studies demonstrated that the detection and characterization of CTCs can facilitate the early diagnosis of relapse or metastasis and improve the early detection and appropriate treatment decisions of various cancers. However, the frequency of CTCs in biofluids is very low (fewer than 10 cells/mL), even in metastatic conditions, and it varies significantly between different types of cancer [79,224]. This low concentration makes their enrichment and subsequent characterization challenging.

CTCs can be found in a patient’s body fluids, either as individual cells or in cell aggregates. Recent data suggest that CTCs may enhance their metastatic potential through homotypic clustering and heterotypic interactions with immune and stromal cells [223]. CTC clusters have been observed in the bloodstream, often associated with non-malignant cells such as white blood cells (WBCs) [225,226]. In most cases, these CTC clusters involve neutrophils [227]. This association between neutrophils and CTCs promotes cell cycle progression within the bloodstream, enhancing the metastatic potential of CTCs. The presence of CTC-WBC clusters has been suggested as an indication of a poor prognosis in cancer patients [227,228]. From a disease pathology perspective, CTCs reflect the metastatic potential of epithelial tumor cells. Additionally, through an epithelial-to-mesenchymal transition (EMT), CTCs can acquire stem cell-like or mesenchymal phenotypes [229].

Currently, the identification and isolation of CTCs from biofluids primarily rely on the presence or absence of specific cell-surface epithelial markers or biophysical properties such as size and deformability [230]. Initially, markers such as the epithelial cell adhesion molecule (EpCAM) and cytokeratins (CKs) were used to detect and isolate CTCs in peripheral blood or bone marrow through IHC or reverse transcription polymerase chain reaction (RT-PCR) [222]. For example, the FDA-approved CellSearch System (Veridex, Warren, NJ, USA) utilizes antibodies against EpCAM and cytokines to detect and enrich CTCs in the peripheral blood of patients with various types of cancers [80,222]. More recently, biomarkers such as the estrogen receptor (ER), human epidermal growth factor receptor 2 (HER2), immune-checkpoint genes, EMT markers, and cancer stem cells (CSCs) have emerged as important markers of CTCs with metastatic potential [222]. Historically, CTCs were initially detected in the peripheral blood of patients with various cancers including breast and prostate cancers. More recently, CTCs have been detected in CNS tumors as well using immunocytochemical and clonogenic assay techniques [78,231,232]. More recently, studies have detected glioma CTCs in both the peripheral blood and CSF of GBM patients, indicating that brain tumor cells can cross the BBB and enter systemic circulation [233,234]. As a result, the use of liquid biopsies is becoming more prominent in the field of GBM. However, because tumor cells from high-grade gliomas such as GBM tend to assume a mesenchymal phenotype rather than an epithelial one, traditional methods of detecting and enriching CTCs from brain tumors using the CellSearch system are not particularly effective [78]. Because of this, several strategies have been explored to detect CTCs in the blood of GBM patients, including targeting glial fibrillary acidic protein (GFAP) with antibodies and the amplification of the *EGFR* gene [235]. Furthermore, the release of CTCs is associated with *EGFR* gene amplification, indicating the growth potential of these cells [235]. CTCs isolated from the blood of GBM patients using various techniques are characterized for *EGFR* amplification. In gliomas, especially GBM, several alterations of the *EGFR* gene have been identified, including amplifications, deletions, and single nucleotide polymorphisms (SNPs). The dysregulation of *EGFR* family members has been linked to the onset and progression of GBM. However, *EGFR* amplification is only found in a subset of GBM cases, making it less effective in isolating CTCs from other GBM subtypes. As discussed extensively in the literature [236], numerous alterations have been documented in gliomas, with certain variants specifically associated with GBM. For example, *EGFR* amplification rates in grade II, III, and IV astrocytomas are 0–4%, 0–33%, and 34–64% respectively. *EGFR* overexpression, indicating increased gene transcription independent of DNA alterations, ranges from 6% to 28%, from 27% to 70%, and from 22% to 89% in grade II, III, and IV astrocytomas, respectively. *EGFRvIII*, characterized by an intragenic deletion spanning exons 2 to 7 affecting the extracellular domain, is closely associated with *EGFR* gene amplification, primarily found in high-grade astrocytomas like GBMs, with occasional exceptions. Various other *EGFR* mutations have been reported, such as carboxy-terminal *EGFR* intracellular domain deletions in exons 25–27, 27–28, and 25–28; *EGFR* gene fusion in GBM involving intron 9 of *SEPT14* or *PSPH*, with an intact tyrosine kinase domain, with or without gene amplification; *EGFR* hypermethylation in early clonal evolution and recurrence; two SNPs in introns 4 and 13 associated with a higher glioma risk; 7p11.2 SNPs; the *EGFR* variant featuring the abnormal splicing of exon 4; *EGFR*c958 deletion spanning amino acids 521–603 in conjunction with *EGFR* amplification; and various specific *EGFR* point mutations.

Additionally, a method that detects increased telomerase activity exclusively in tumor cells has been used to detect and enrich CTCs derived from brain tumors [237]. This approach employs an adenoviral detection system that has proven effective in detecting CTCs in patients with brain tumors. Notably, clinical data indicate that the adenoviral detection-based system can distinguish between pseudoprogression and actual tumor progression in CNS malignancies [237]. Recent data demonstrated that CTCs can be identified in the blood of patients with seven different subtypes of brain glioma by examining the aneuploidy of chromosome 8 (CEP8-FISH) [238]. Furthermore, a novel microfluidic device, the CTC-iChip, was used to efficiently detect and enrich CTCs using a panel of mesenchymal gene expression signatures (referred to as STEAM: *SOX2, Tubulin beta-3, EGFR, A2B5,* and *c-MET*) in the peripheral blood of GBM patients [239]. This device achieves this by selectively removing leukocytes [239]. Although not yet clinically tested, proteoglycans (i.e., complex molecules consisting of a protein core and a sugar side chain) have also been used to detect and enrich CTCs in several types of cancers. For example, in high-grade gliomas, recombinant malaria VAR2CSA protein rVAR2, which targets the proteoglycan chondroitin, has been used to detect CTCs in the blood by binding to tumor-specific oncofetal chondroitin sulfate [233]. The recurrence rate and the progression of low-grade gliomas have been associated with the presence of CTCs that exhibit a CSC-like phenotype, rendering them a promising target for detecting CTCs [240]. Further evidence from recent data indicates that CTCs can also be detected in the blood of pediatric patients with brain tumors [43]. Table 7 presents examples of ongoing clinical studies investigating the potential of CTCs as diagnostic, predictive, and prognostic biomarkers in GBM.

Furthermore, the ability to detect and isolate viable CTCs from patients with different cancer types, including brain tumors, has facilitated the development of CTC-derived cell lines, xenografts, and 3D models such as spheroids and organoids for functional studies such as drug testing, among others [79,241,242,243]. In summary, CTCs collected from patients with CNS tumors including GBM provide valuable models for studying molecular alterations specific to CNS tumors. These models can help monitor tumor progression and guide the development of targeted therapies [244].

## 9. Circulating Protein Profiling as a Potential Biomarker

Proteins found in circulation, originating either from tumors or the immune system, play a vital role in the development and progression of various cancer types [245]. The cancer secretome, which comprises all proteins secreted or shed by cancer cells into the extracellular compartment or bodily fluids, promotes cancer progression metastasis [246,247]. Cancer-secreted proteins, such as enzymes, cytokines, and growth factors, are involved in various biological and physiological processes, including immune responses and cell–cell communication. Many of these secreted proteins are present in measurable amounts in blood and other bodily fluids including CSF, urine, and others, making them potential biomarkers that are more accessible than proteins within tumor tissue. Various proteomic approaches have been used to analyze the cancer cell secretome such as mass spectrometry-based (label-based and label-free), antibody-based, and bead-based array methods including ELISA, Western blotting techniques, as well as gel-based methods such as two-dimensional polyacrylamide gel electrophoresis (2D), among others [248].

Circulating proteins, such as those found in blood (i.e., serum or plasma), CSF, and urine, have shown promise in guiding diagnosis, assessing treatment responses, and understanding mechanisms of treatment resistance. According to recent literature [56], assessing the correlation between tumor pathology and protein expression including circulating protein markers provides new opportunities for identifying key pathways, novel biomarkers, the risk of cancer risk assessment, and therapeutic targets for cancer prevention across different cancer types [249,250].

Previous research indicated that, in cancer patients, the increased secretion of proteins can lead to higher levels of circulating proteins in various biological fluids, including blood [251]. However, the high degree of differences in the concentrations of abundant proteins and circulating proteins secreted in the blood can limit the detection and clinical utility of circulating proteins [249,251].

Currently, various tumor-specific protein biomarkers are routinely used in clinical practice, including PSA, CEA, CA15-3, CA125, CA19-9, CYFRA21-1, S100, NSE, ProGRP, sHER2, SCCA, HE-4, and CA72-4, spanning various cancer types. Additionally, the recent proteomic profiling of various cancer patients for circulating proteins has identified other potential circulating tumor protein biomarkers such as thymidine kinases, DNAse activity, circulating nucleosomes, soluble receptors of advanced glycation end products (sRAGE), high-mobility group box 1 (HMGB1), and immunogenic cell death markers [252,253]. A recent study has reported [250] the identification of novel therapeutic targets for cancer among 2074 circulating proteins and the risk of nine cancers. Another study [254] has identified proteomic risk factors for cancer using prospective and exome analyses of 1463 circulating proteins and the risk of 19 cancers in the UK biobank and identified 618 proteins. Of these, 107 persist for cases diagnosed more than seven years after blood collection. A total of 29 of 618 were associated with genetic analyses, and 4 had support from a long time-to-diagnosis (>7 years).

As for CNS malignancies, various efforts have been undertaken to discover circulating proteins specific to CNS malignancies. For example, Kikuchi et al. [255] were the first group to report blood-based protein biomarkers in brain tumors. The findings from this study indicated elevated levels of immunosuppressive acidic proteins, including alpha-1 antitrypsin and alpha-1 acidic glycoprotein, as well as endothelial cell-derived thrombomodulin and glycoprotein fibronectin in glioma patients compared to those in both non-glioma individuals and healthy subjects [255]. Furthermore, several studies demonstrated that angiogenesis-related proteins such as VEGF, soluble endothelial growth factor receptor-1 (sVEGFR-1), and basic fibroblast growth factor (FGF-2) were elevated in circulation in patients with various glioma grades [256,257,258]. Other notable circulating protein biomarkers for brain tumors and metastases include tumor cells’ extracellular matrix remodeling proteins, including matrix metalloproteinases (MMPs) and tissue inhibitors of metalloproteinases (TIMPs), contributing to tumor classification according to staging [259]. Other examples include the detection of increased plasma levels of interleukins 2 (IL-2) and its receptor, tumor necrosis factor-alpha (TNFα), tumor necrosis factor beta (TNFβ), neural cell adhesion molecule (NCAM), neuropeptide Y (NPY), and chitinase-3-like protein 1 (CHI3L1/YKL-40), which have diagnostic significance in CNS malignancies [260,261,262]. Notably, CHI3L1/YKL-40 demonstrates potential as a prognostic biomarker for grade 4 glioma, showing an inverse association with overall survival [261,263]. Plasminogen activator inhibitor-1 (PAI-1) presents another marker of interest, with its serum levels showing a negative correlation with the progression-free survival (PFS) of brain tumor patients [263].

Furthermore, circulating proteins can play a role not only in early diagnosis and prognosis but also in monitoring the effectiveness of cancer treatments. For instance, patients with recurrent high-grade glioma treated with bevacizumab, rather than cytotoxic agents, exhibited increased plasma MMP2 protein levels after eight weeks, correlating with enhancements in both PFS and overall survival (OS) [264].

The profiling of circulating proteins as potential biomarkers for GBM offers several advantages. They provide a non-invasive alternative to traditional tissue biopsies, reducing patient discomfort and risk. These proteins can detect glioblastoma early and monitor tumor progression or the response to therapy in real time. Protein biomarkers can reflect dynamic changes in the TME, offering specific disease information. Blood tests for circulating proteins are accessible and can be repeated, facilitating regular monitoring. Additionally, the analysis of circulating proteins can help tailor personalized treatment strategies based on an individual’s tumor profile and provide insights into the heterogeneous nature of glioblastomas.

However, there are also disadvantages. The lack of standardized methodologies for blood collection, processing, and analysis affects data reliability. Circulating protein levels can be influenced by various unrelated factors, leading to false positives or variability in the results. Some proteins may not be specific to glioblastoma and could be elevated in other conditions, complicating the interpretation. Tumor-related proteins, especially those specific to GBM, might be present at low levels, requiring highly sensitive detection techniques. The measurement of these proteins necessitates advanced, costly, and technically demanding technologies.

In summary, circulating proteins hold great promise as non-invasive or minimally invasive biomarkers for GBM, and several proteomic profiling studies of patient samples have identified potential circulating tumor-specific protein biomarkers in blood or CSF, significantly improving our understanding of disease initiation, progression, and therapy responses. However, several technical and biological challenges need to be addressed to fully realize their potential in clinical practice. Additionally, further investigation is needed to determine their translational significance through extensive clinical validation before they can be reliably used in clinical settings.

## 10. Circulating Metabolomic and Lipidomic Profiling as a Potential Biomarker

As highlighted in the literature [56,265,266,267], liquid biopsy-based metabolome profiling has also been utilized to identify and quantify various compounds in the biofluids of cancer patients, including the blood, urine, CSF, and others in patients with various cancers. Metabolomics, like proteomics, genomics, and other omics technologies, has enabled the discovery of novel biomarkers and improved our understanding of disease imitation, progression, and responses to therapy. In addition, like genomics or proteomics, metabolomics provides insights into the various aspects of molecular mechanisms associated with disease. This underscores its relevance in disease pathogenesis and development, offering a dynamic, comprehensive, and precise depiction of the disease phenotype. Circulating metabolites, such as amino acids, carbohydrates, nucleosides, nucleotides, lipids, vitamins, and fatty acids, among many others, primarily contribute to cellular structure maintenance and signal transduction via secondary messenger molecules [268,269].

Metabolic changes in brain tumors disrupt both anabolic and catabolic processes, impacting cellular signaling pathways and functions, which contribute to tumor development and treatment resistance over time [270,271]. For example, a study analyzing plasma and saliva samples from 159 newly diagnosed high-grade brain tumor GBM patients found significant associations between methionine and arginine, associated with a better prognosis, while kynurenine levels, an intermediate of tryptophan metabolism, were correlated with poor survival outcomes [270,271]. Similarly, another study reported significant differences in uridine and ornithine levels between low- and high-grade glioma patients [272]. Additionally, the metabolite profiles of patients with primary grade 4 gliomas showed higher concentrations of antioxidant compounds like α-tocopherol and γ-tocopherol, suggesting their potential as biomarkers of high-grade brain tumor progression [273].

Due to the abundant presence of lipids in the brain and their crucial roles in functions such as forming lipid membranes, energy metabolism, and signal transmission, lipidomics has emerged as a specialized field within brain tumor metabolomics research, may provide insights into disease pathogenesis and development, and can be used as a biomarker for disease diagnosis and prognosis and for monitoring therapy responses [267,274,275]. Although studies on brain tumor-specific lipid biomarkers using liquid biopsy approaches are limited, recent research identified candidate diagnostic biomarkers and proposed new opportunities for lipid biomarker identification. In contrast to other neurological conditions, few studies have employed a liquid biopsy approach to identify and characterize lipid biomarkers specific to brain tumors. For example, a recent study has identified [265] 11 plasma lipids as potential diagnostic indicators for malignant brain tumors. Furthermore, a comprehensive lipid profiling study explored the complexity of fatty acids encompassing 99 treatment-naive GBMs [276,277]. The authors analyzed 582 lipid species from 75 tumors and compared them with 7 normal brain tissues. The findings from the study revealed more than 500 distinct lipid species [276,277]. The study revealed that the lipidome of brain tumors varies according to the *IDH* status and tumor molecular subtype, distinguishing them from normal brain tissues. For example, the mesenchymal GBM subtype showed an increase in the overall levels of glycerolipids (e.g., triacylglycerols) and a decrease in glycerophospholipids. In contrast, the proneural subtype exhibited enrichment in very long-chain fatty acids (VLCFAs) and glycerophospholipids with polyunsaturated fatty acid (PUFA) side chains [276]. Another recent study investigated the potential of blood lipids as biomarkers for the diagnosis of GBM by using an unbiased lipidomic approach and identified differentially regulated lipid species, including fatty acids, glycerolipids, glycerophospholipids, saccharolipids, sphingolipids, and sterol lipids, between patients with GBM and controls [278].

Additionally, studies have shown that the CSF of glioma patients contains differentially enriched metabolites such as lactic acid, malic acid, succinate, and phosphoenolpyruvate compared to that of control patients, with variations observed based on the *IDH* status of GBM patients [279]. Overall, recent progress in metabolite detection techniques and the integration of predictive models into lipid biomarker discovery workflows present new opportunities for biomarker discovery efforts [280].

## 11. Circulating Extracellular Vesicle (EV) and Exosome Profiling as a Potential Biomarker

Recent literature [56] has highlighted the potential of liquid biopsy-based extracellular vesicles (EVs, comprising both microvesicles and exosomes) [270] profiling as a biomarker across various cancers. EVs, a lipid-bilayer-bound organelle, released into the extracellular space by both tumor and healthy cells, offer several advantages as analytes compared to other liquid biopsy-derived substrates [281]. EVs are broadly classified into exosomes, microvesicles, and apoptotic bodies based on their size, morphology, and method of generation [282,283]. EVs contain a range of biomolecules like DNA, RNA (e.g., coding and non-coding RNAs such as miRNAs), proteins, metabolites, lipids, etc. and are present in various biofluids such as serum, plasma, CSF, urine, saliva, ascites, semen, breast milk, ocular samples, tears, nasal lavage fluid, and synovial fluid [284,285,286,287,288,289,290]. Under physiological conditions, tumor cells generate EVs at a significantly higher rate compared to normal cells [291]. Furthermore, EVs and exosomes contain RNAs such as mRNAs and miRNAs, DNA, proteins, and lipids, all shielded from enzymatic degradation [46]. This composition offers a more accurate representation of biological processes compared to ctDNA or other biomarkers. Importantly, EVs carry markers on their surface specific to their parental cell of origin, aiding in predicting organotropic metastases [292]. Additionally, EVs demonstrate a higher frequency of detecting cancer mutations in the DNA they contain compared to ctDNA [293,294].

As highlighted in recent studies, EVs are implicated in tumor progression and propagation by promoting various cellular processes such as proliferation, extracellular matrix remodeling, angiogenesis, immune modulation and evasion, and metastasis [295,296,297,298]. The quantitative analysis of EVs originating from tumor cells has been linked to prognosis in several cancer types [299]. It has been shown that tumor-derived EVs contain double-stranded genomic DNAs that have the same mutations specific to the tumor, including *KRAS, EGFR, BRAF*, and *TP53* [300,301,302].

Studies have also investigated EVs from brain tumor patients, revealing specific genetic mutations such as *EGFR* gene mutations in DNA isolated from serum-derived EVs in GBM patients [296] and miRNA (e.g., miR-320 and miR-574-3p) and noncoding RNA (RNU6-1) expression patterns distinct from healthy individuals [303]. The study identified significant associations between the levels of miR-574-3p and miR-320, RNU6-1, and the diagnosis of GBM. Notably, *RNU6-1* emerged as a consistent independent predictor of GBM diagnosis [303]. The study has also demonstrated that higher levels of a panel of surface markers on EVs can differentiate different subtypes of brain tumors [303]. Furthermore, prior research has indicated that GBM patients with elevated levels of tumor-associated molecules such as podoplanin and *EGFRvIII* within EVs exhibit low response rates to chemoradiation therapy, which typically involve radiotherapy in combination with temozolomide [304]. Additionally, post-resection tumor recurrence generally correlates with a rise in EV levels in the plasma of individuals with CNS malignancies [305].

Moreover, EVs and exosomes derived from high-grade gliomas have been demonstrated to promote tumor growth and neoangiogenesis, indicating the potential for exosomes to facilitate metastasis [306]. Additionally, these exosomes mirror the hypoxic condition of glioma cells and contribute to the hypoxia-induced activation of vascular cells during tumor progression [306]. Likewise, the analysis of proteomic profiles of serum EVs from patients with histologically defined medulloblastoma has revealed their potential involvement in cancer cell proliferation and migration [307]. The findings from this study suggest the tumor-suppressive activity of the transcription factor hepatocyte nuclear factor 4 alpha (HNF4A) [307].

Overall, EV-based liquid biopsy offers a promising avenue for biomarker identification and therapeutic response monitoring in CNS malignancies [281]. Table 8 and Table 9 present examples of ongoing clinical studies investigating the potential of EVs and exosomes as diagnostic, predictive, and prognostic biomarkers in GBM. However, as summarized in Table 10, EVs have challenges that can be used as biomarkers, and further research in large and diverse cohorts is necessary to enhance sensitivity, specificity, and reproducibility before clinical implementation.

## 12. Challenges and Future Perspectives

Due to the high mortality rate associated with GBM, there is an urgent need for minimally invasive approaches to better inform the prognosis of the disease and to monitor treatment responses. Current diagnoses of GBM rely on radiological imaging by MRI and tumor tissue data; however, there are some challenges and limitations with these approaches. As summarized in Table 10, traditional MRI is a non-invasive technique that allows for the visualization of brain tumor anatomy and can guide surgery. However, it cannot distinguish between high-grade gliomas [308], concurrent pathological processes, and other brain disorders and does not differentiate true progression from pseudoprogression, which is an important clinical challenge, making imaging findings challenging to interpret [308]. Although the advanced MRI techniques and amino acid PET imaging may address some of these shortcomings, the correlation between MRI findings and GBM molecular alterations remains unclear. Thus, tissue biopsy remains necessary for comprehensive GBM diagnosis. However, tissue biopsies pose various limitations. For example, tumor tissue biopsies are invasive and cannot be repeated easily, and they may not adequately represent the entire tumor. Similarly, they cannot provide the real-time assessment of tumor activity.

To address these challenges, the liquid biopsy-based analysis of circulating biomarkers has been explored as an alternative or complementary approach to conventional techniques for GBM diagnosis, disease progress, and therapy response monitoring. As highlighted in the literature [270], liquid biopsies offer distinct advantages over current methods, such as their non-invasive nature and simple procedure and their ability to repeatedly sample throughout treatment without invasive procedures. Moreover, high-grade tumors may lead to an increased permeability of the BBB, facilitating molecular transport [309], suggesting that liquid biopsies might detect tumor-related information before clinical progression occurs [310,311]. However, tissue biopsies offer more accurate insights into tumor morphology and the microenvironment. Therefore, the objective of a liquid biopsy should be to complement and enhance the diagnostic accuracy and monitoring of GBM patients to monitor dynamic changes in the tumor throughout the therapy period by providing additional information alongside tissue biopsies.

Currently, there are no clinically validated circulating biomarkers for GBM management, and this is in part due to the BBB, which poses a challenge by restricting the transportation of various types of macromolecules between the blood and the brain, contributing to the scarcity of circulating biomarkers in this context. In addition, each type of circulating biomarker, such as proteins, ctDNAs, miRNAs, CTCs, EVs, and others, exhibits distinct advantages and limitations, as summarized in Table 10. For example, monitoring patients longitudinally through serial sampling to detect specific tumor mutations and DNA methylation changes in ctDNAs could offer valuable insights into tumor behavior and treatment resistance. However, the detection rates of ctDNA in GBM patients vary widely (10–55%), underscoring the need for comprehensive studies with larger cohorts to understand its role in GBM.

It is important to note that most circulating biomarkers have a short half-life in the blood [46,98], although some are protected within EVs like microvesicles and exosomes, shielding them from degradation [46]. For instance, ctDNAs typically have a short half-life in the blood, ranging from 16 min to 2.5 h [117], primarily released by cells undergoing necrosis or apoptosis [42,46,98,312]. In contrast, mRNA has a longer half-life of approximately 16.4 h, while the half-life of miRNA in the blood is approximately 16.42 ± 4.2 h [313]. Similarly, cytokines, peptides, and other proteins typically have half-lives of less than 1 h in the blood [314]. Likewise, CTCs exhibit a short half-life of approximately 1–2.4 h [315,316]. Similarly, EVs of most cell types have a half-life of 1–30 min [317]. Biodistribution studies reveal that EVs have a half-life of less than 30 min in vivo across most tissues [318]. Exosomes, a subtype of EVs, also have a short half-life in circulation, lasting approximately 2–30 min [319], with up to 90% being removed within 5 min after infusion [320]. They are mainly taken up by macrophages associated with the organs of the mononuclear phagocyte systems such as the liver, lungs, and spleen [321]. The stability of metabolites varies depending on the type; for example, L-arginine has a plasma half-life of approximately 6 ± 2 (range 3.7–8.4) minutes [322], while S-Adenosyl methionine persists for about 100 min [323]. Similarly, L-kynurenine, an intermediate in tryptophan metabolism associated with poor survival outcomes in GBM patients, has a plasma half-life of approximately 94 min [324].

Despite the challenges involving detecting and profiling circulating factors as biomarkers, these circulating biomarkers, whether they are ctDNAs, miRNAs, CTCs, EVs or exosomes, metabolites, proteins, or a combination of these, have the potential to complement the current methodologies for managing GBM patients. Additionally, the longitudinal collection of samples at multiple time points could allow for monitoring tumor progression or detecting pseudoprogression in a minimally invasive manner. When comparing different biofluid sources, such as ctDNA in blood versus CSF, CSF appears to be more representative due to its proximity to the brain. However, collecting CSF is far more invasive and potentially carries more risk than blood collection.

Although variations in miRNA expression levels throughout the disease course, including before and after treatment or at recurrence, offer valuable insights for managing GBM, a better understanding of the mechanisms behind miRNA dysregulation in GBM pathogenesis might provide promising opportunities for developing targeted therapies and overcoming therapeutic resistance. However, the clinical utility of miRNAs as circulating biomarkers is hampered by small cohort sizes and a lack of standardized methodologies for blood collection, RNA extraction, and sequencing. Additionally, miRNAs tend to have a lower specificity compared to ctDNA. Therefore, larger prospective studies need to be conducted to validate the utility of miRNAs as diagnostic, prognostic, and therapy response biomarkers for GBM. The specific miRNAs that can be used as biomarkers in the early screening stages of GBM are still under investigation [4,5,26,57,59,78,82,121,133,154,161,170,210,249,263,266,286,288,325,326,327,328,329,330,331,332,333,334,335,336,337,338,339,340,341,342,343,344,345,346,347,348,349,350,351,352,353,354].

For example, unlike many other cancers, the utility of CTCs as a diagnostic tool in GBM is limited, as by the time clinical symptoms manifest and diagnosis is made, the disease is often in an advanced stage. Additionally, technical limitations hinder the isolation and quantitation of CTCs in GBM diagnosis. While some studies have detected CTCs in GBM, indicating their potential to breach the BBB, these findings require validation through larger studies. Only a few studies have investigated GBM-derived CTCs, with detection rates ranging from 20 to 77% in GBM patients; however, detection rates vary depending on the techniques employed for CTC isolation. For example, one of the initial reports [355] identifying CTC clusters in GBM indicates the potential for GBM clusters to cross the BBB. This discovery holds substantial clinical importance and necessitates larger cohort studies to validate its reproducibility. In this study, Krol et al. [355] investigated the presence and composition of CTCs at various intervals in 13 patients with progressive GBM participating in an open-label phase 1/2a trial involving the microtubule inhibitor BAL101553. Notably, CTC clusters consisting of 2 to 23 cells were identified in serially collected samples in a GBM patient exhibiting pleomorphism and extensive necrosis, spanning disease progression [355]. The exome sequencing of these GBM CTC clusters revealed variants in 58 cancer-associated genes, including *ATM*, *PMS2*, *POLE*, *APC*, *XPO1*, *TFRC*, *JAK2*, *ERBB4*, and *ALK* [355]. Although CTCs can provide insights into the protein, DNA, and RNA levels and can be used for functional assays such as drug testing, organoid cultures, or xenograft models, CTCs are rare, with a frequency of about 1 cell per 10^9^ blood cells [43,46,81,312]. In addition, CTCs may only represent part of the tumor’s heterogeneity, and the process of isolating them is challenging.

Similarly, although the emerging field of EVs and exosomes in GBM has shown promise, with studies detecting the EGFRvIII deletion variant in tumor tissue (39.5%) matching with *EGFRvIII* expression in exosomes (44.7%) and correlating it with poor survival, mirroring findings in exosomes, EVs and exosomes come with certain challenges and limitations. For example, the heterogeneity of EVs and exosomes, the technical challenges associated with their collection and characterization, and the small cohort sizes used for their collection and analysis require further research in this field. In addition, although EVs and exosomes carry proteins, DNA, RNA, and miRNA, they can be released by all cells, including tumor cells, and current isolation methods can introduce contaminants, creating obstacles in the analysis and use of EVs and exosomes [46,356,357].

As highlighted in a recent review [78] and summarized in Table 10, each type of circulating marker—whether ctDNA, miRNAs, CTCs, exosomes, or others—presents distinct advantages and disadvantages. 

Therefore, a combination of markers carrying predictive markers like *IDH1*, *MGMT*, and *EGFRvIII* might be beneficial for non-invasive diagnostic and prognostic assessments.

Because of these challenges, improving the technologies for the consistent isolation and characterization of these biomarkers is needed. In addition, the integration of better computational tools such as machine learning and artificial intelligence for large sets of data emerging from these omics-based profiling efforts in future research is warranted. Furthermore, conducting large-scale clinical studies to explore these biomarkers with longitudinal clinical correlations to assess their impact on clinical outcomes is required for their clinical application. Nevertheless, liquid biopsy holds significant promise in managing GBM patients.

Despite recent progress, the process of identifying and validating biomarkers remains a major challenge in biomedical research. Recent progress in various “omics” fields, coupled with bioinformatics, biostatistics, machine learning (ML) algorithms, and artificial intelligence (AI), has expedited the discovery and development of drugs and biomarkers [325,358,359].

As highlighted in recent literature, including our own contributions [325,358,359,360], the integration of ML and AI technologies for the analysis and interpretation of large and complex omics datasets has emerged as a central focus in accelerating various domains of biomedical research. This encompasses drug discovery and development, diagnostic imaging, and the analysis of genomic and other multiomics data, including various types of circulating biomarkers, consequently enhancing decision-making processes significantly. As a result, ML and AI technologies are poised to become integral components of routine large-scale omics data analysis, incorporating multi-omics data such as genomics, transcriptomics, proteomics, metabolomics, circulating biomarkers, and others, providing deeper insights into disease mechanisms and drug responses. The growing interest among academia, industry, healthcare providers, regulatory agencies, and other stakeholders underscores the momentum behind developing AI-driven approaches to analyzing and interpreting complex biological datasets.

As a result, improving the technologies for biomarker profiling and characterization and tools for analyzing the complex and large-scale omics datasets will further enable correlations of complex and multi-factorial clinical data with therapies targeted to specific molecular alterations and tailor treatment strategies based on individual molecular profiles. These approaches will also enable the identification of novel biomarkers associated with prognosis and predicting responses to treatment in GBM, elucidating the underlying sensitivity and resistance mechanisms to therapy and the identification of promising targets for the therapy of GBM to guide treatment decisions [338,361,362]. Furthermore, future large-scale clinical studies evaluating novel biomarker data and the complex and multi-factorial clinical data with therapies targeted to specific molecular mechanisms are warranted to assess their impact on clinical outcomes.

In summary, the integration of ML and AI technologies alongside profiling patient biofluids, including those with GBM for circulating biomarkers, is expected to expedite data analysis and assist in interpreting the complex large datasets generated from these efforts. This advancement holds promise for enhancing diagnostics, prognostics, and therapy monitoring, thereby improving decision-making processes and ultimately leading to better patient outcomes.

## 13. Conclusions

Various liquid biopsy techniques are currently used to detect diagnostic, predictive, and prognostic markers in biofluids, monitor treatment responses, and improve patient outcomes. This approach is especially significant for CNS tumors, supplementing traditional MRI or CT scans in the ongoing monitoring of patients following initial diagnosis and treatment. This approach holds particular importance for CNS tumors, complementing the traditional MRI or CT scans for monitoring patients following initial diagnosis and treatment. Liquid biopsy for CNS malignancies generally involves analyzing blood (serum or plasma), CSF, the vitreous body, and urine for detecting and profiling diverse biomarkers (e.g., ctDNA, miRNA, CTCs, EVs, proteins, metabolites, and others), showing promise in diagnosing and managing CNS tumors [56,151]. The idea that liquid biopsy-based tests can be used to screen many cancer types has huge potential. The utility of liquid biopsy for cancer screenings is still in its infancy, although some recent data presented at ASCO 2024 suggest that a lack of ctDNA in blood correlates with favorable patient outcomes [363]. The team from the Breast Cancer Now Toby Robins Research Center at the Institute of Cancer Research (ICR), London, demonstrated that the NeXT Personal liquid biopsy test can detect ctDNA at any point after surgery or during the follow-up period [363]. An analysis from the ChemoNEAR study showed that the presence of ctDNA was associated with a high risk of future relapse and poorer overall survival [363]. Molecular residual disease was detected in all 11 patients who relapsed (1 June, ASCO 2024) in a 76-patient study [363].

Although liquid biopsy has progressed in other cancer types, its application in CNS tumors is still in development and is not used as part of the SoC in neuro-oncology. In addition, further research is needed to overcome technological hurdles and deepen our understanding of brain tumor biology through liquid biopsy, which is critical to harnessing its potential for brain tumor diagnosis and management. Furthermore, gaining insights into brain tumor biology, such as understanding the role of CTCs and EVs, the frequency of genetic mutations in ctDNA, and the influence of biomolecules like miRNAs on transcriptome, proteome, and metabolome profiles, and of the aberrant expression and altered levels of ctDNAs, miRNAs, proteins, metabolites, and EVs on disease pathogenesis, progression, and responses to therapies is essential for advancing technology and assay development. Thus, overcoming technological challenges and enhancing our understanding of CNS tumor biology through liquid biopsy will be crucial in realizing its potential for CNS tumor diagnosis and management. To achieve this, attention needs to be directed towards increasing the availability of liquid biopsy specimens to researchers, developing standardized sample collection methods, enhancing the specificity and sensitivity of tumor-associated signal detection, and employing tailored downstream analytical and bioinformatics techniques such as incorporating ML and AI systems into the data analysis process.

Ultimately, the implementation of successful brain tumor-focused liquid biopsy efforts requires a coordinated collaboration among academic and clinical research labs, industry, and global organizations such as the International Brain Research Organization (IBRO) and the American Brain Tumor Association (ABTA), among others, which can establish brain-tumor-specific liquid biopsy entities like already existing initiatives such as BloodPac, Cancer-ID, PANCAID, European Liquid Biopsy Society (ELBS), Liquid Biopsy Consortium, International Society of Liquid Biopsy (ISLB), etc. While these efforts will be resource-intensive, their clinical implementation is expected to be cost-effective in the long run.

## Figures and Tables

**Figure 1 ijms-25-07974-f001:**
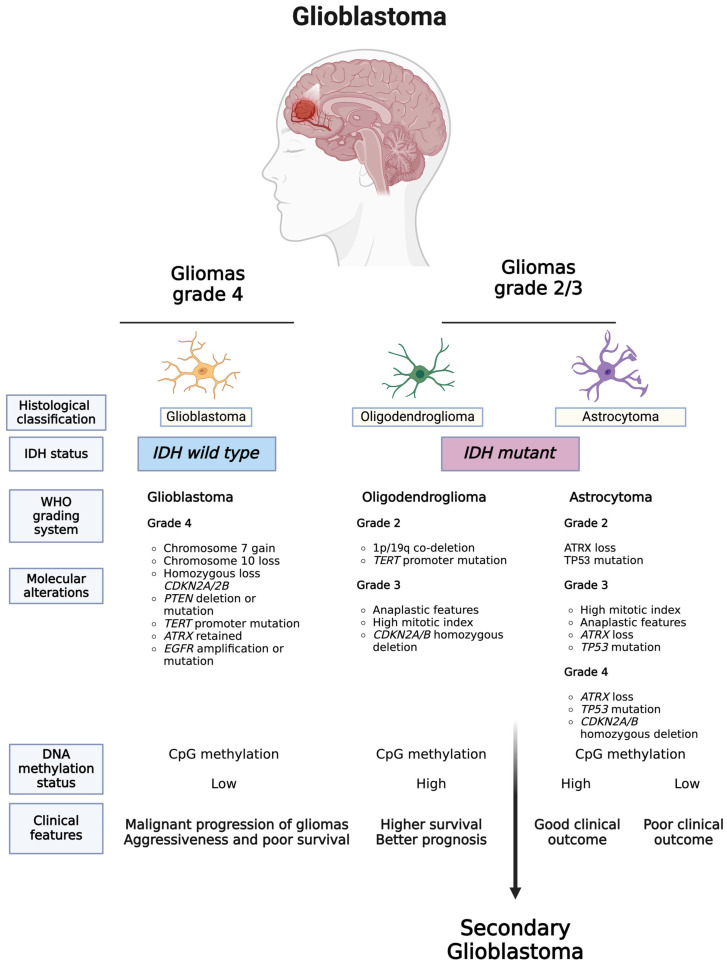
Updated WHO classification of tumors of the CNS. The 2021 WHO classification of CNS tumors introduced significant changes, such as limiting the diagnosis of GBM to only *IDH* wild type tumors, reclassifying previously diagnosed *IDH*-mutated GBMs as astrocytomas, *IDH*-mutated, and grade 4, and requiring the presence of *IDH* mutations for tumors to be classified as astrocytomas or oligodendrogliomas. For the abbreviations, go to the abbreviations list at the end of the text. Created with BioRender.com (accessed on 6 March 2023).

**Figure 2 ijms-25-07974-f002:**
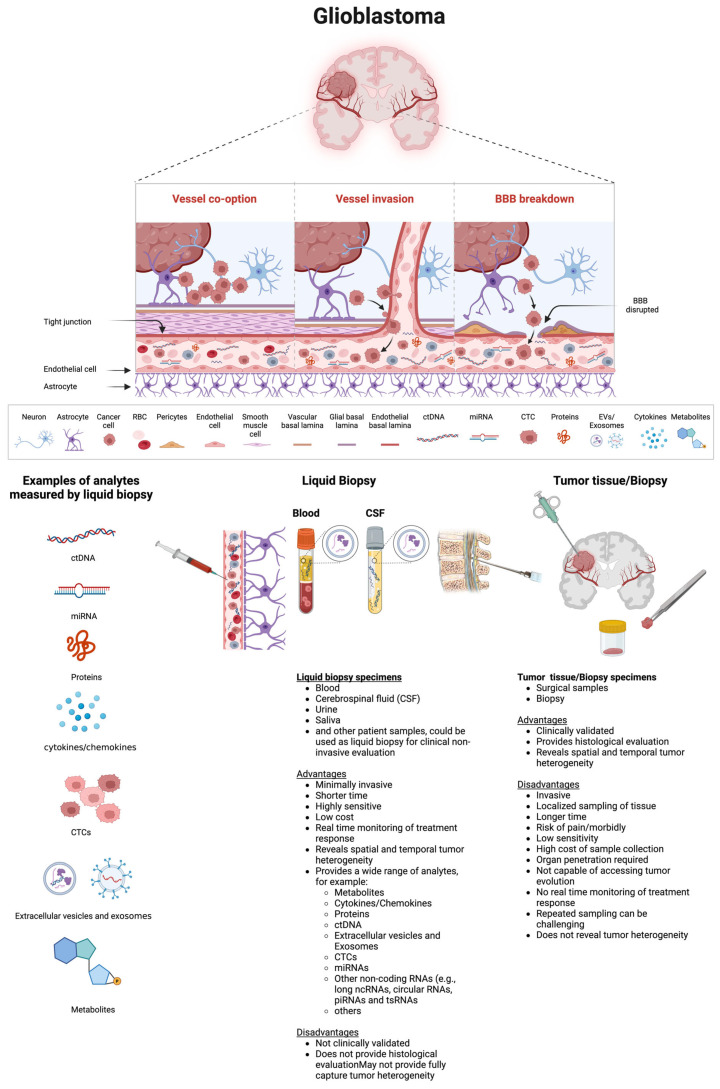
Examples of biomarkers measured in circulation, along with the advantages and disadvantages of liquid biopsy versus tumor tissue biopsy, are shown. This schematic representation illustrates circulating biomarkers released from the tumor into the bloodstream through the partially disrupted BBB. These biomarkers may also be directly secreted into the CSF. In patients with GBM, a compromised BBB allows circulating biomarkers such as ctDNAs, miRNAs, EVs, CTCs, proteins, and metabolites to enter the bloodstream or CSF. These biomarkers can be collected through blood or CSF draws and subsequently analyzed. The illustration provides a breakdown of tumoral components within the circulatory system. Various analytical methods, including PCR, qRT-PCR, NGS, WGS, immunoaffinity capture, ELISA, mass spectrometry, chemiluminescent immunoassay, and density gradient centrifugation, have been used to detect circulating analytes. Each circulating analyte can be assessed for tumor-specific changes such as various types of mutations, epigenetic modifications, DNA fragmentation patterns, nucleosome patterning, chromosomal aberrations, and the presence, absence, or changes in levels of ctDNAs, miRNAs (and other noncoding RNAs as well as mRNAs), CTCs, proteins, cytokines, metabolites, EVs, or exosomes, along with post-translational modifications. Each type of biomarker detection method, whether blood- or CSF-based or tissue-based, has unique advantages and disadvantages in diagnosing and monitoring GBM patients. Abbreviations: BBB, blood–brain barrier; CSF, cerebrospinal fluid; CTCs, circulating tumor cells; ctDNA, circulating tumor DNA; DNA, deoxyribonucleic acid; EVs, extracellular vesicles; NGS, next-generation sequencing; PCR, polymerase chain reaction; RNA, ribonucleic acid. Created with BioRender.com (accessed on 11 July 2024).

**Figure 3 ijms-25-07974-f003:**
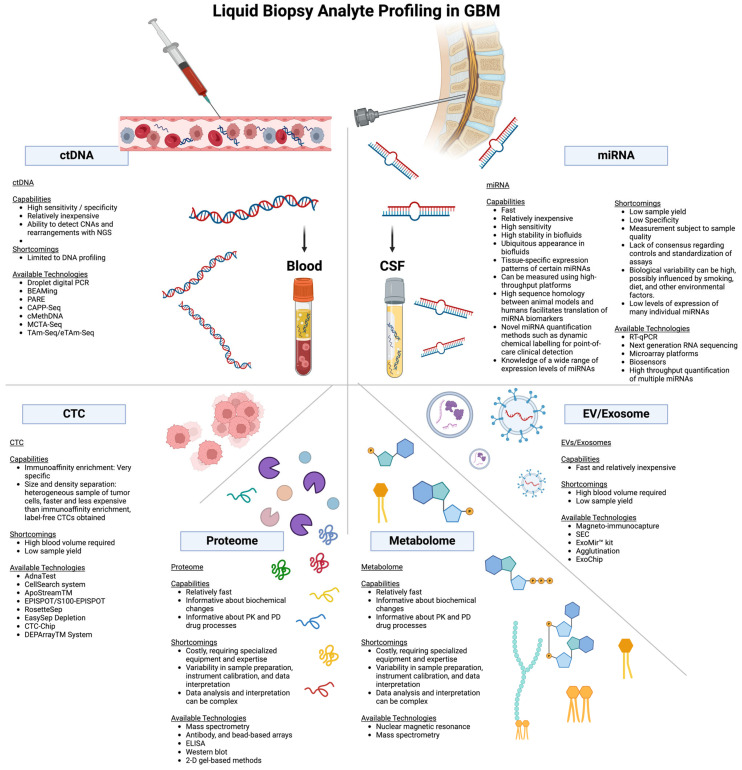
Comparison of examples of liquid biopsy techniques, highlighting their capabilities, shortcomings, and available technologies. Abbreviations: CNA, copy number alterations; CTC, circulating tumor cells; ctDNA, circulating tumor DNA; DNA, deoxyribonucleic acid; ddPCR, droplet digital PCR; miRNA, microRNA; NGS, next-generation sequencing; PCR, polymerase chain reaction; RNA, ribonucleic acid. Created with BioRender.com (accessed on 11 July 2024).

**Table 1 ijms-25-07974-t001:** Recent clinical studies that evaluated liquid biopsy as a potential biomarker in GBM. A search was conducted on Clinicaltrials.gov using the terms “liquid biopsy” and “Glioblastoma” on 6 March 2024.

Rank	NCT Number	Title	Status	Study Results	Conditions	Interventions	Phases	Study Type	URL
1	NCT05383872	Blood-Brain Barrier Disruption (BBBD) for Liquid Biopsy in Subjects With Glioblastoma Brain Tumors	Recruiting	No Results Available	Glioblastoma, Glioma, Liquid Biopsy	Device: Focused Ultrasound (Exablate Model 4000, Insightec Ltd. Tirat Carmel, Israel)	Not Applicable	Interventional	https://ClinicalTrials.gov/show/NCT05383872
2	NCT05099068	Profiling Program of Cancer Patients with Sequential Tumor and Liquid Biopsies (PLANET)	Recruiting	No Results Available	Advanced/Metastic Solid Tumors, Glioblastoma, Chronic Leukemia Lymphocytic	Biological: Blood and tumor samples	Not Applicable	Interventional	https://ClinicalTrials.gov/show/NCT05099068
3	NCT04776980	Multimodality MRI and Liquid Biopsy in GBM	Withdrawn	No Results Available	Glioblastoma Multiforme, Brain Tumor, Adult: Glioblastoma, Brain Tumor, Recurrent, Brain Tumor, Primary	Diagnostic Test: Post Feraheme Infusion MRI	Early Phase 1	Interventional	https://ClinicalTrials.gov/show/NCT04776980
4	NCT05695976	GRETeL: Tumor Response to Standard Radiotherapy and TMZ Patients With GBM	Recruiting	No Results Available	Glioblastoma, Glioma, Malignant			Observational	https://ClinicalTrials.gov/show/NCT05695976
5	NCT05934630	Testing Cerebrospinal Fluid for Cell-free Tumor DNA in Children, Adolescents, and Young Adults with Brain Tumors	Active, not recruiting	No Results Available	Anaplastic Astrocytoma, Diffuse Brainstem Glioma, Glioblastoma Multiforme, High-grade Astrocytoma NOS, Fibrillary Astrocytoma, Low-Grade Astrocytoma, Nos, Pilocytic Astrocytoma, Choroid Plexus Carcinoma, CNS Primary Tumor, Nos, Atypical Teratoid/Rhabdoid Tumor, Medulloblastoma, Supratentorial Primitive Neuroectodermal Tumor, Ependymoma, NOS, Anaplastic Oligodendroglioma, Oligodendroglioma, Nos, CNS Germ Cell Tumor, Pineoblastoma, Diffuse Leptomeningeal Glioneuronal Tumor			Observational	https://ClinicalTrials.gov/show/NCT05934630
6	NCT05281731	Sonobiopsy for Noninvasive and Sensitive Detection of Glioblastoma	Recruiting	No Results Available	Glioblastoma, Glioblastoma Multiforme	Device: Sonobiopsy, Procedure: Research blood, Genetic: Cancer Personalized Profiling, Device: Definity^®^ microbubbles, Lantheus, Inc. N. Billerica, MA	Not Applicable	Interventional	https://ClinicalTrials.gov/show/NCT05281731
7	NCT06136611	Preoperative Preradiotherapy TTFields	Not yet recruiting	No Results Available	Glioblastoma	Device: TTFields	Not Applicable	Interventional	https://ClinicalTrials.gov/show/NCT06136611
8	NCT00463008	Pharmacologic Study of Methotrexate in Patients Undergoing Stereotactic Biopsy for Recurrent High-Grade Glioma	Completed	No Results Available	Brain and Central Nervous System Tumors	Drug: methotrexate, Other: pharmacological study	Not Applicable	Interventional	https://ClinicalTrials.gov/show/NCT00463008

**Table 4 ijms-25-07974-t004:** Recent clinical studies that evaluated miRNAs as potential biomarkers in GBM.

Biomarker	Study Title	Cancer Types	Patients (*n*) (Cases/Controls)	Controls	Biofluid	Method	Alterations	Results	References
miR-7	Dynamic expression of 11 miRNAs in 83 consecutive primary and corresponding recurrent glioblastoma: correlation to treatment, time to recurrence, overall survival and *MGMT* methylation status	GBM	83	Recurrent GBM vs primary GBM	FFPE tissues	qRT-PCR	Downregulation	Significant change in the expression of miR-7, miR-9, miR-21, miR-26b, mirR-124a, miR-199a, and let-7f in the recurrent tumor compared to the primary tumor. In the recurrent tumor, miR-15b, let-7d, and let-7f significantly changed compared to both treatment options.	[183]
miR-10b	Human glioma growth is controlled by microRNA-10b	GBM	258	NA	Tissue	TCGA tumor tissue data analysis	Upregulation	miR-10b is upregulated in both low-grade and high-grade gliomas. High miR-10 levels associated with much shorter patient survival compared with the low miR-10 expressors. miR-10b expression correlated with the expression of genes that belong to “G1/S transition”, “G2/M transition”, “S phase”, and “M phase of mitotic cell cycle” bioterms.	[184,185]
miR-15b miR-21	miR-15b and miR-21 as Circulating Biomarkers for Diagnosis of Glioma	GBM	16/30	Neurologic disorders	Serum	qRT-PCR	Downregulation	Elevated miR-15b and miR-21 were detected in the blood of GBM patients. miR-15b and miR-21 showed high sensitivity (90%) and specificity (100%) in distinguishing glioma patients from non-glioma patients	[174]
miR-15b miR-23a miR-133a miR-150 miR-197 miR-497 miR-548b-5p	Identification of seven serum microRNAs from a genome-wide serum microRNA expression profile as potential noninvasive biomarkers for malignant astrocytoma	Newly diagnosed Astrocytomas (WHO grades III-IV)	33/80	Healthy	Serum	qRT-PCR	Downregulation	Seven miRNAs were significantly decreased in grades II–IV patients (*p* < 0.001), and all seven miRNA panels exhibited a high sensitivity (88.00%) and specificity (97.87%) in predicting malignant astrocytomas and were downregulated in tumor tissues compared to normal tissues. Furthermore, these miRNAs in serum were markedly elevated after operation (*p* < 0.001).	[173]
miR-21 miR-128 miR-342-3p	Plasma specific miRNAs as predictive biomarkers for diagnosis and prognosis of glioma	GBM	10/10	Healthy	Plasma	qRT-PCR	miR-21 Upregulation miR-128 Downregulation miR-342-3 Downregulation	miR-21, miR-128, and miR-342-3p were significantly altered in GBM patients compared to normal controls but not in other brain tumors like meningioma or pituitary adenoma. Additionally, miR-128 and miR-342-3p were positively correlated with histopathological grades of glioma.	[171]
miR-30c		GBM	53	53 paired GBM tissues and adjacent normal brain tissues	Tissue	qRT-PCR	Downregulation	The expression of miR-30c was significantly downregulated in GBM tissues compared to in normal tissues. The miR-30c decrease was more pronounced in high-grade GBM tissues compared to in low-grade tissues.	[186,187]
miR-125b-2	MicroRNA-125b-2 confers human glioblastoma stem cells resistance to temozolomide through the mitochondrial pathway of apoptosis	GBM	N/A	Normal human brain tissues obtained from patients with severe TBI who needed post-trauma surgery	Tissue	qRT-PCR	Downregulation	miR-125b-2 expression was upregulated in GBM tissues and the corresponding stem cells (GBMSCs), which conferred resistance to TMZ.	[188]
miR-128	Serum microRNA-128 as a biomarker for diagnosis of glioma	GBM	61/53	Healthy	Serum	qRT-PCR	Downregulation	The expression of miR-128 was notably reduced in preoperative glioma serum compared to both normal controls and meningioma serum samples (both *p* < 0.001). After surgery, miR-128 expression significantly increased (*p* < 0.001), but it did not reach normal levels (*p* < 0.001). Additionally, low miR-128 levels in serum and tissue were associated with a high pathological grade and low KPS	[172]
miR-128 miR-342-3p	A specific miRNA signature in the peripheral blood of glioblastoma patients	GBM	20/20	Healthy	Blood	qRT-PCR	miR-128 Upregulation miR-342-3 Downregulation	Among 1158 tested miRNAs, 52 exhibited significant deregulations.Only miR-128 (upregulated) and miR-342-3p (downregulated) remained significant after correction for multiple testing.	[181]
miR-130b	MicroRNA-130b promotes cell proliferation and invasion by inhibiting peroxisome proliferator-activated receptor-γ in human glioma cells	Astrocytic gliomas	12	4 on-neoplastic brain specimens as controls	Fresh tissues	qRT-PCR	Upregulation	The expression level of miR-130b was found to be markedly higher in human glioma tissues than in non-neoplastic brain specimens	[189]
miR-181c miR-181d	Clinical Relevance and Interplay between miRNAs in Influencing Glioblastoma Multiforme Prognosis	GBM	112		FFPE	qRT-PCR	Downregulation	The OS curves show that the combination of low miR-648 and miR-181c or miR-181d expressions is associated with a worse prognosis	[190]
miR-181c miR-181d miR-21 miR-195 miR-196b miR-648 miR-767.3	Identification of *MGMT* Downregulation Induced by miRNA in Glioblastoma and Possible Effect on Temozolomide Sensitivity	GBM	112		FFPE	qRT-PCR	Upregulation Downregulation	miR-21 and miR-196b were upregulated and miR-767.3 was downregulated in GBM. Low expression of miR-181c, miR-195, miR-648, and miR-767.3p was associated with positive *MGMT* IHC. A significant association was found between unmethylated cases and the low expression of miR-181d and miR-648 and between methylated cases and the low expression of miR-196b. Negative *MGMT* IHC; in methylated patients and in the cases with miR-21, miR-196b was associated with a better OS.	[191]
miR-182	Potential Diagnostic and Prognostic Value of Plasma Circulating MicroRNA-182 in Human Glioma	GBM	39/54	Healthy	Plasma	qRT-PCR	Upregulation	miR-182 in glioma patients was higher than that in healthy controls, which was significantly associated with the KPS score and WHO grade.	[178]
miR-183	Up-regulation of microRNA-183 promotes cell proliferation and invasion in glioma by directly targeting *NEFL*	GBM	44	44 human astrocytoma samples and 20 normal brain tissues	Tissue	QRT-PCR	Upregulation	miR-183 was significantly upregulated in astrocytoma tissues and glioblastoma cell lines. NEFL as a novel target gene of miR-183. The expression levels of NEFL are inversely correlated with those of miR-183 in human astrocytoma clinical specimens	[192]
miR-203	MALAT1 is a prognostic factor in glioblastoma multiforme and induces chemoresistance to temozolomide through suppressing miR-203 and promoting thymidylate synthase expression	GBM (TMZ-resistant and non-resistant patients)	192	96 patients showing a response (CR and PR) to TMZ treatment and 96 patients showing no response (SD and PD).	FFPE tissues and serum	qRT-PCR	Downregulation	miR-203 was downregulated by lncRNA *MALAT1*. LncRNA MALAT1 inhibition re-sensitized TMZ-resistant cells through upregulating miR-203 and downregulating TS expression	[193]
miR-205	Downregulation of serum microRNA-205 as a potential diagnostic and prognostic biomarker for human glioma	GBM	27/45	Healthy	Serum	qRT-PCR	Downregulation	Serum miR-205 levels were significantly lower in patients with glioma than in healthy controls and demonstrated a stepwise decrease with ascending pathological grades and KPS scores. Higher miR-205 serum levels were correlated with a longer OS.	[179]
miR-210	Serum microRNA-210 as a potential noninvasive biomarker for the diagnosis and prognosis of glioma	GBM	42/50	Healthy	Serum	qRT-PCR	Upregulation		[175]
miR-221 miR-222	Clinical impact of circulating oncogenic miRNA-221 and MiRNA-222 in glioblastoma multiform	GBM	20/20	Healthy	Serum	qRT-PCR	Upregulation	miR-221 and -222 were significantly increased in GBM cases as compared to healthy individuals. Higher levels of miR-221 and miR-222 were correlated with PD and patients with a worse PFS and OS.	[182]
miR-320a	miR-320a functions as a suppressor for gliomas by targeting *SND1* and *β-catenin*, and predicts the prognosis of patients	Astrocytic gliomas	120	Surgical specimens of 120 astrocytic gliomas and 20 nontumoral brain tissues	FFPE tissues	ISH	Downregulation	miR-320a expression was decreased in human glioma tissues and cell lines. Reduced miR-320a expression was inversely correlated with glioma grades and Ki-67 indexes but positively correlated with patients’ survival.	[194]
miR-339-5p miR-21-5p mR-92b-3p miR-182-5p	Simultaneous miRNA and mRNA transcriptome profiling of glioblastoma samples reveals a novel set of OncomiR candidates and their target genes	GBM	50	50 GBM tissue samples and 7 healthy tissue samples	Tissue samples	qRT-PCR	Upregulation	miR-339-5p, miR-21-5p, miR-92b-3p, and miR-182-5p were found to be significantly upregulated in GBM samples. An increased miR-21 expression level was correlated with an older age at diagnosis in GBM.	[195]
miR-454-3p	Plasma miR-454-3p as a potential prognostic indicator in human glioma	GBM	22/70	Healthy	Plasma	qRT-PCR	Upregulation	The levels of miR-454-3p were higher in high-grade gliomas than in low-grade gliomas. The post-operative plasma levels of miR-454-3p were downregulated significantly compared to the pre-operative levels. High miR-454-3p levels are associated with a poorer prognosis.	[176]
miR-497	A restricted signature of serum miRNAs distinguishes glioblastoma from lower grade gliomas	GBM	10/15	Healthy	Serum	qRT-PCR	Downregulation	miR-497 and miR-125b serum levels were decreased depending on tumor stages, with reduced levels in GBM than in lower-grade tumors.	[177]
miR-519a	miR-519a enhances chemo sensitivity and promotes autophagy in glioblastoma by targeting *STAT3/Bcl2* signaling pathway	GBM	48	24 patients with recurrent GBM treated with TMZ before the second surgery and 24 patients with primary GBM without TMZ treatment	FFPE		Downregulation	Downregulation of miR-519a and upregulation of STAT3 in recurrent GBM tissues were detected compared to primary GBM tissues.	[196,197]
miR-595	MiR-595 targeting regulation of *SOX7* expression promoted cell proliferation of human glioblastoma	GBM	8	8 paired human GBM tissues and the matched tumor-adjacent tissues	Tissue	QRT-PCR	Upregulation	MiR-595 expression was significantly upregulated in GBM tissues and cells.	[198,199]
miR-758-5p	Mir-758-5p Suppresses Glioblastoma Proliferation, Migration and Invasion by Targeting *ZBTB20*	GBM	55	55 paired GBM tissues and adjacent normal tissues	Tissue samples	qRT-PCR	Downregulated	miR-758-5p was significantly downregulated in GBM tissues. High miR-758-5p expression indicated an enhanced prognosis of patients with GBM.	[200]
miR-146b-5p	miR-146b-5p functions as a tumor suppressor by targeting *TRAF6* and predicts the prognosis of human gliomas	Astrocytic gliomas	147	20 nontumoral brain tissues as controls	FFPE tissues	ISH	Downregulation	Reduced miR-146b-5p expression was inversely correlated with the grades and Ki-67 index in 147 human glioma specimens but positively correlated with patients’ survival.	[1]

Abbreviations: CR, Complete response; FFPE, Formalin-Fixed Paraffin-Embedded; ISH, in situ hybridization; KPS, Karnofsky Performance Status; lncRNA, long noncoding RNA; miR, microRNA; N/A, not available; NEFL, neurofilament light polypeptide; OS, overall survival; PD, progressive disease; PFS, progression-free survival; PR, partial response; qRT-PCR, quantitative reverse transcription polymerase chain reaction. SD, stable disease; TBI, traumatic brain injury; TS, thymidylate synthase.

**Table 5 ijms-25-07974-t005:** Recent preclinical studies that evaluated oncomirs and tumor-suppressive miRNAs and their functions in GBM.

miRNA	Type of miRNA	Expression	Targets	Functional Assay	Tumor Grade	Sources	References
miR-10b	Oncomir	Upregulation	*BCL2L11/BIM*, *RhoC*, *uPAR*	Cell proliferation, invasion, cell cycle, cell death	Grade III–IV glioma	GBM tissue	[185,201]
miR-17-92 cluster	Oncomir	Upregulation	*TGFβRII*, *SMAD4*, *CTGF*, *CAMTA1*, *POLD2*	Cell viability, proliferation, apoptosis, angiogenesis	Grade III–IV glioma	Cell line, GBM tissue	[202]
miR-21	Oncomir	Upregulation	*CASP3*, *CASP9*, *STAT3*	Cell apoptosis	Grade III–IV	Cell line, GBM tissue	[203,204]
miR-23a	Oncomir	Upregulation	*HOXD10*, *uPAR*, *RhoA*, *RhoC*	Cell invasion	GBM glioma	Cell line	[205]
miR-92b-3p	Oncomir	Upregulation	*TGFBRII*, *SMAD4*, *CAMTA1*	Cell migration, invasion, apoptosis	GBM	GBM tissue	[195,206,207]
miR-182-5p/21-5p/339-5p	Oncomir	Upregulation	*Ras*, *HIF-1*, *MAPK*	Cell migration, invasion, apoptosis	GBM	Cell line, GBM tissue	[195]
miR-183	Oncomir	Upregulation	*NEFL*	Cell proliferation	GBM	Cell line, GBM tissue	[192]
miR-296-5p	Oncomir	Upregulation	*HMGA1*, *CASP8*	Cell invasion, glioma stem cells	GBM	Xenografts	[208,209]
miR-595	Oncomir	Upregulation	*Sox7*	Cell proliferation, aggressiveness, migration	GBM	Cell line, GBM tissue	[198,199]
miR-1290	Oncomir	Upregulation	*SOCS4*	Cell proliferation, migration, invasion, chemoradiotherapy resistance	GBM	Cell line	[210]
miR-7	Tumor-suppressive miR	Downregulation	*FAK*	Cell invasion, migration	Grade III–IV glioma	Cell line, GBM tissue	[211]
miR-30c	Tumor-suppressive miR	Downregulation	*Sox9*	Cell proliferation, migration, invasion	GBM	Cell line, GBM tissue	[186,187]
miR-124/137	Tumor-suppressive miR	Downregulation	*CDK6*	Cell cycle, proliferation	Grade III–IV glioma, GSCs	GSCs, GBM tissue	[212]
miR-125b	Tumor-suppressive miR	Downregulation	*CDK6*, *CDC25A*, *MMP2/9*	Cell invasion, cell cycle, apoptosis, stemness, resistance to TMZ	Grade III–IV glioma	GSCs, GBM tissue	[188]
miR-181	Tumor-suppressive miR	Downregulation	*Bcl-2*, *CCNB1*	Cell proliferation, apoptosis, invasion, angiogenesis, radio-chemosensitivity	Grade III–IV glioma	Cell line, GBM tissue	[213,214]
miR-451	Tumor-suppressive miR	Downregulation	*CAB39*, *PI3K/Akt/SNAI1*	Cell proliferation, apoptosis, cell cycle, EMT	GBM and GSCs	Cell lines, xenograft	[215]
miR-490	Tumor-suppressive miR	Downregulation	*TERF2*, *TNKS2*, *SMG1*	Cell proliferation, telomere maintenance	GBM	Cell line	[216]
miR-519a	Tumor-suppressive miR	Downregulation	*STAT3*, *Bcl-2*	Cell proliferation, migration, invasion, apoptosis	GBM	Cell line, GBM tissue, xenografts	[196,197]
miR-758-5p	Tumor-suppressive miR	Downregulation	*ZBTB20*	Cell migration, invasion, proliferation	GBM	Cell line, GBM tissue, xenografts	[200]

Abbreviations: *Akt*: AKT serine/threonine kinase, *Bcl-2*: B-cell lymphoma 2, *BCL2L11/BIM*: BCL2 like 11, *CAB39*: calcium binding protein 39, *CAMTA1*: calmodulin binding transcription activator 1, *CASP3*: caspase 3, *CASP8*: caspase 8, *CASP9*: caspase 9, *CCNB1*: cyclin B1, *FAK*: focal adhesion kinase, *CDK6*: cyclin-dependent kinase 6, *CDC25A*: cell division cycle 25A, CTGF: connective tissue growth factor, GSCs: glioma stem cells, HIF-1: hypoxia-inducible factor 1, HMGA1: high mobility group AT-hook 1, *HOXD10*: homeobox D10, *MAPK*: mitogen activated kinase-like protein, *NEFL*: neurofilament light chain, *MMP2/9*: matrix metalloproteinase 2/9, *PI3K*: phosphatidylinositol-4,5-bisphosphate 3-kinase, *POLD2*: DNA polymerase delta 2, accessory subunit, *RhoA*: Ras homolog family member A, *RhoC*: Ras homolog family member C, *SMAD4*: SMAD family member 4, *SNAI1*: snail family transcriptional repressor 1, *SOCS4*: suppressor of cytokine signaling 4, *Sox7*: SRY-box transcription factor 7, *Sox9:* SRY-box transcription factor 9, *STAT3*: signal transducer and activator of transcription 3, *TERF2*: telomeric repeat-binding factor 2, *TGFBRII*: transforming growth factor-beta receptor type 2, *TNKS2*: tankyrase 2, *SMG1*: SMG1 nonsense mediated mRNA decay-associated PI3K-related kinase, *uPAR*: urokinase-type plasminogen activator receptor, *ZBTB20*: zinc finger and BTB domain containing 20.

**Table 6 ijms-25-07974-t006:** Recent clinical studies that evaluated miRNAs as potential biomarkers in GBM. A search was conducted on Clinicaltrials.gov using the terms “miRNA”, “microRNA”, and “Glioblastoma” on 6 March 2024.

Rank	NCT Number	Title	Status	Study Results	Conditions	Interventions	Phases	Study Type	URL
1	NCT01849952	Evaluating the Expression Levels of MicroRNA-10b in Patients with Gliomas	Recruiting	No Results Available	Astrocytoma, Oligodendroglioma, Oligoastrocytoma, Anaplastic Astrocytoma, Anaplastic Oligodendroglioma, Anaplastic Oligoastrocytoma, Glioblastoma, Brain Tumors, Brain Cancer			Observational	https://ClinicalTrials.gov/show/NCT01849952
2	NCT03866109	A Study Evaluating Temferon in Patients with Glioblastoma & Unmethylated *MGMT*	Recruiting	No Results Available	Glioblastoma Multiforme	Drug: Temferon	Phase 1, Phase 2	Interventional	https://ClinicalTrials.gov/show/NCT03866109
3	NCT05328089	Vacuolar ATPase and Drug Resistance of High-Grade Gliomas	Recruiting	No Results Available	Glioblastoma Multiforme			Observational	https://ClinicalTrials.gov/show/NCT05328089
4	NCT02751138	Determination of Immune Phenotype in Glioblastoma Patients	Completed	No Results Available	Glioblastoma Multiforme	Procedure: Surgery		Observational	https://ClinicalTrials.gov/show/NCT02751138
5	NCT02544178	Study of Neurological Complication After Radiotherapy for High Grade Glioblastoma	Unknown status	No Results Available	Leukoencephalopathy	Radiation: Brain radiotherapy		Observational	https://ClinicalTrials.gov/show/NCT02544178
6	NCT06203496	Monitoring of Patients with Diffuse Gliomas Using Circulating miRNAs	Recruiting	No Results Available	Glioma, Malignant	Diagnostic Test: Blood sample		Observational	https://ClinicalTrials.gov/show/NCT06203496
7	NCT03630861	Establishment of a Signature of Circulating microRNA as a Tool to Aid Diagnosis of Primary Brain Tumors in Adults	Completed	No Results Available	Brain Tumors			Observational	https://ClinicalTrials.gov/show/NCT03630861
8	NCT03770468	Molecular Genetic, Host-derived and Clinical Determinants of Long-term Survival in Glioblastoma	Active, not recruiting	No Results Available	Glioblastoma	Procedure: Blood drawl		Observational	https://ClinicalTrials.gov/show/NCT03770468
9	NCT03025893	A Phase II/III Study of High-dose, Intermittent Sunitinib in Patients with Recurrent Glioblastoma Multiforme	Unknown status	No Results Available	Glioblastoma Multiforme, Glioblastoma, Adult, Glioblastoma, Recurrent Brain Tumor, GBM	Drug: Sunitinib, Drug: Lomustine	Phase 2, Phase 3	Interventional	https://ClinicalTrials.gov/show/NCT03025893
10	NCT05871021	Protective VEGF Inhibition for Isotoxic Dose Escalation in Glioblastoma	Not yet recruiting	No Results Available	Glioblastoma	Radiation: Dose escalation of radiation dose beyond the therapeutic standard	Phase 2	Interventional	https://ClinicalTrials.gov/show/NCT05871021

**Table 7 ijms-25-07974-t007:** Recent clinical studies that evaluated CTCs as potential biomarkers in GBM. A search was conducted on Clinicaltrials.gov using the terms “circulating tumor cells”, “CTCs”, and “Glioblastoma” on 6 March 2024.

Rank	NCT Number	Title	Status	Study Results	Conditions	Interventions	Phases	Study Type	URL
1	NCT03861598	Carvedilol With Chemotherapy in Second Line Glioblastoma and Response of Circulating Tumor Cells	Terminated	No Results Available	Glioblastoma Multiforme, Glioblastoma	Drug: Carvedilol	Early Phase 1	Interventional	https://ClinicalTrials.gov/show/NCT03861598
2	NCT03980249	Anti-Cancer Effects of Carvedilol with Standard Treatment in Glioblastoma and Response of Peripheral Glioma Circulating Tumor Cells	Withdrawn	No Results Available	Glioblastoma, Glioblastoma Multiforme	Drug: Carvedilol	Early Phase 1	Interventional	https://ClinicalTrials.gov/show/NCT03980249
3	NCT01135875	Laboratory Study of Early Tumor Markers in the Peripheral Blood of Glioblastoma Multiforme Patients	Completed	No Results Available	Glioblastoma Multiforme, Healthy Volunteers	Other: GBM Patients, Other: Normal Controls		Observational	https://ClinicalTrials.gov/show/NCT01135875
4	NCT00001148	Detecting Malignant Brain Tumor Cells in the Bloodstream During Surgery to Remove the Tumor	Completed	No Results Available	Astrocytoma, Glioblastoma, Glioma			Observational	https://ClinicalTrials.gov/show/NCT00001148
5	NCT03115138	Evaluation of Circulating Tumor DNA as a Theranostic Marker in the Management of Glioblastomas.	Terminated	No Results Available	Glioblastoma, Molecular Disease	Other: Correlation between molecular anomalies of the primary tumor and circulating tumor DNA	Not Applicable	Interventional	https://ClinicalTrials.gov/show/NCT03115138
6	NCT04776980	Multimodality MRI and Liquid Biopsy in GBM	Withdrawn	No Results Available	Glioblastoma Multiforme, Brain Tumor, Adult: Glioblastoma, Brain Tumor, Recurrent, Brain Tumor, Primary	Diagnostic Test: Post-Feraheme Infusion MRI	Early Phase 1	Interventional	https://ClinicalTrials.gov/show/NCT04776980
7	NCT05695976	GRETeL: Tumor Response to Standard Radiotherapy and TMZ Patients With GBM	Recruiting	No Results Available	Glioblastoma, Glioma, Malignant			Observational	https://ClinicalTrials.gov/show/NCT05695976
8	NCT02669173	Capecitabine + Bevacizumab in Patients with Recurrent Glioblastoma	Active, not recruiting	No Results Available	Glioblastoma	Drug: Capecitabine, Drug: Bevacizumab	Phase 1	Interventional	https://ClinicalTrials.gov/show/NCT02669173
9	NCT00905060	HSPPC-96 Vaccine with Temozolomide in Patients with Newly Diagnosed GBM	Completed	Has Results	Brain and Central Nervous System Tumors	Biological: HSPPC-96, Drug: Temozolomide, Procedure: Standard Surgical Resection	Phase 2	Interventional	https://ClinicalTrials.gov/show/NCT00905060
10	NCT00621686	Bevacizumab and Sorafenib in Treating Patients with Recurrent Glioblastoma Multiforme	Completed	Has Results	Brain and Central Nervous System Tumors	Biological: Bevacizumab, Drug: sorafenib tosylate	Phase 2	Interventional	https://ClinicalTrials.gov/show/NCT00621686

**Table 8 ijms-25-07974-t008:** Recent clinical trials that use EVs in GBM. A search was conducted on Clinicaltrials.gov using the terms “extracellular vesicles” and “Glioblastoma” on 6 March 2024.

Rank	NCT Number	Title	Status	Study Results	Conditions	Interventions	Phases	Study Type	URL
1	NCT03576612	GMCI, Nivolumab, and Radiation Therapy in Treating Patients with Newly Diagnosed High-Grade Gliomas	Active, not recruiting	No Results Available	Glioma, Malignant	Biological: AdV-tk, Drug: Valacyclovir, Radiation: Radiation, Drug: Temozolomide, Biological: Nivolumab, Other: Laboratory Biomarker Analysis	Phase 1	Interventional	https://ClinicalTrials.gov/show/NCT03576612

**Table 9 ijms-25-07974-t009:** Recent clinical studies that evaluated exosomes as potential biomarkers in GBM. A search was conducted on Clinicaltrials.gov using the terms “exosomes” and “Glioblastoma” on 6 March 2024.

Rank	NCT Number	Title	Status	Study Results	Conditions	Interventions	Phases	Study Type	URL
1	NCT05328089	Vacuolar ATPase and Drug Resistance of High-Grade Gliomas	Recruiting	No Results Available	Glioblastoma Multiforme			Observational	https://ClinicalTrials.gov/show/NCT05328089
2	NCT05864534	Phase 2a Immune Modulation with Ultrasound for Newly Diagnosed Glioblastoma	Recruiting	No Results Available	Newly Diagnosed Glioblastoma, Glioblastoma, Isocitric Dehydrogenase (*IDH*)-Wildtype, Gliosarcoma, Glioblastoma Multiforme	Drug: Balstilimab, Drug: Botensilimab, Drug: Liposomal Doxorubicin, Device: Sonocloud-9 (SC-9)	Phase 2	Interventional	https://ClinicalTrials.gov/show/NCT05864534
3	NCT05698524	A Study of Temodar with Abexinostat (PCI-24781) for Patients with Recurrent Glioma	Recruiting	No Results Available	Recurrent High-Grade Glioma, Anaplastic Astrocytoma, Anaplastic Oligodendroglioma, Glioblastoma, Gliosarcoma	Drug: PCI 24781, Drug: Temozolomide	Phase 1	Interventional	https://ClinicalTrials.gov/show/NCT05698524

**Table 10 ijms-25-07974-t010:** Summary of the advantages and disadvantages of each diagnostic method, including MRI, tissue biopsy, and various liquid biopsy-based methods.

Diagnostic Method	Advantages	Disadvantages
MRI	Noninvasive procedure with no known risk Provides initial diagnosis and anatomic characterization of GBM through a non-invasive procedure	Challenges in distinguishing GBM from other brain diseases and concurrent pathological processes Challenges in correlating MRI features with molecular characteristics Challenges in differentiating true tumor recurrence from PsP
Tissue biopsy	Clinically validated Provides histological evaluation Enables histologic and molecular characterization of the tumor Reveals spatial and temporal tumor heterogeneity	Highly invasive procedure with associated risks Localized sampling of tissue Longer time to complete the procedure Low sensitivity Limited or no repeated sampling Fails to accurately reveal intra-tumoral heterogeneity Unable to assess tumor activity in real time Organ penetration required Cannot reveal tumor evolution Lacks real-time monitoring of treatment response. High cost of sample collection
ctDNA	Higher levels than those of CTCs High specificity ctDNA quantity correlates with tumor burden and disease stage Easier to collect and established detection techniques available Ability to detect CNAs and rearrangements with NGS Relatively inexpensive	ctDNA level varies depending on the tissue type and cancer stage Gliomas have the lowest detectable levels of ctDNA ctDNA concentration in cancer is very low (180 ng/mL) and potentially even lower in GBM ctDNA has a short half-life (<2.5 h) Released mainly by apoptotic or necrotic cells and therefore represents only a subpopulation of tumor cells Sensitivity of detection limited
miRNAs	Relative high stability in biofluids (half-life ~16.4 h) Fast High sensitivity Ubiquitous appearance in biofluids Tissue-specific expression patterns of certain miRNAs Can be measured using high-throughput platforms High sequence homology between animal models and humans facilitates translation of miRNA biomarkers Novel miRNA quantification methods such as dynamic chemical labeling for point-of-care clinical detection Knowledge of a wide range of expression levels of miRNAs Relatively inexpensive	No standardized methods for RNA extraction and sequencingLess specific than ctDNA Low sample yield Low specificity Measurement subject to sample quality Lack of consensus regarding controls and standardization of assays Biological variability can be high, possibly influenced by smoking, diet, and other environmental factors Low levels of expression of many individual miRNAs
CTCs	Highly specific Offer insights into protein, DNA, and RNA levels Immunoaffinity enrichment: highly specific Size and density separation: heterogeneous sample of tumor cells, faster and less expensive than immunoaffinity enrichment, label-free CTCs obtained	High blood volume required Low sample yield Lack of standardized methods for isolating and characterizing CTCsLow presence in blood CTCs have a short half-life (1–2.4 h)May not represent the whole tumor
EVs/Exosomes	Fast and relatively inexpensive Carry RNAs, proteins, and lipids, all of which are protected from enzyme degradation Able to cross an intact BBB Released by both normal and cancer cells	Lack of standardized methods for isolating EVs EVs are highly heterogeneous EVs have a short half-life (<30 min) Exosomes have a short half-life (<30 min) Released by non-neoplastic cells, resulting in a background of non-tumoral EVs in the blood Requires a high volume of blood Produces a low sample yield
Metabolites	Metabolomics enables endogenous metabolite profiling Reveals information about the current pathophysiological status of patients Offers insights into the tumor’s unique biochemical landscape Reveals alterations in the cellular phenotype due to its specific focus on biochemical changes Reveals information about PK and PD drug processes Enables the monitoring of disease progression and treatment response through changes in metabolic and lipidomic profiles	Complex data analysis. The vast amount of data generated requires sophisticated analysis techniques, which can be challenging and time-consuming. Variability in sample preparation, instrument calibration, and data interpretation can affect the reproducibility and accuracy of results Metabolites have a short half-life (<100 min) Can be costly, requiring specialized equipment and expertise Translating findings from metabolic and lipidomic profiling into clinical practice may take time and further validation The heterogeneity of GBM may lead to challenges in interpreting metabolic and lipidomic profiles, as different regions of the tumor may exhibit distinct profiles
Circulating nucleosome-associated histonemodifications	Circulating nucleosome-associated histone modifications are highly stable Can be detected by ELISA and ChLIA Epigenetics is an emerging and intensive field of research	Low specificity
Circulating proteins as potential biomarkers	Circulating proteins can be detected in blood (i.e., serum or plasma), CSF, and urine Circulating proteins are more accessible and can be repeated multiple times, facilitating the regular monitoring of the disease Circulating proteins can potentially detect glioblastoma at an early stage and monitor tumor progression or response to therapy in real time They have shown promise in guiding diagnosis, assessing treatment responses, and understanding mechanisms of treatment resistance Protein biomarkers can reflect the dynamic changes in the tumor microenvironment, providing specific information about the state of the disease Potential for Personalized Medicine: Analysis of circulating proteins can help tailor personalized treatment strategies based on the specific protein expression profile of an individual’s tumor Circulating proteins can provide insights into the heterogeneous nature of glioblastomas, potentially revealing multiple aspects of tumor biology	Variability in sample preparation, instrument calibration, and data interpretation can affect the reproducibility and accuracy of the results Variability and sensitivity. Circulating protein levels can be influenced by various factors unrelated to glioblastoma, such as inflammation, infection, or other comorbidities, leading to potential false positives or variability in the results Limited sensitivity and specificity. Some circulating proteins may not be specific to glioblastoma and could be elevated in other types of cancers or diseases, complicating the interpretation of the results Circulating proteins have a short half-life (<1 h) Some GBM-related proteins might be present at very low levels in the circulation (blood or CSF), making their detection challenging and requiring highly sensitive analytical techniques Advanced and sensitive technologies are required to accurately measure circulating proteins, which can be costly and technically demanding Many potential protein biomarkers for glioblastoma are still in the research phase and require extensive clinical validation before they can be reliably used in a clinical setting Further investigation is needed to determine their translational significance in clinical settings.

Abbreviations: BBB, blood–brain barrier; ChLIA, chemiluminescence immunoassay; CTCs, circulating tumor cells; ctDNA, circulating tumor DNA; ELISA, enzyme-linked immunosorbent assay; EVs, extracellular vesicles; GBM, glioblastoma; PD, pharmacodynamics; PK, pharmacokinetics; miRNAs, microRNAs; MRI, magnetic resonance imaging; PsP, pseudoprogression.

## Data Availability

All data supporting the findings of this study are available within the paper. The data presented in this study are openly available at https://pubmed.ncbi.nlm.nih.gov/ and https://clinicaltrials.gov/, all accessed starting in 1 March 2024.

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
