# Peer review of "Circulating Liquid Biopsy Biomarkers in Glioblastoma: Advances and Challenges"

_ijms, 2024, doi:10.3390/ijms25147974_

Round 1

Reviewer 1 Report

Comments and Suggestions for Authors

Dr. Seyhan writes a comprehensive review of where we stand in terms of liquid biopsy in glioblastoma. The paper is generally well written. The following changes should be considered.

1. In the introduction section, the authors cite a paper showing that mOS of GBM is about 8 months (line 52), and a different papers showing the mOS to be less than 15 months. The former paper is probably "real-world" data and the latter are clinical trials, but the two seem to contradict each other, so a reader not familiar with GBMs may be confused. Please clarify.

2. Many of the paragraphs in the paper are only one sentence. This may be a matter of style, but paragraphs should be multiple sentences which are related to each other. Please correct.

3. The following papers, one showing that liquid biopsy of cyst fluid in cystic brain tumors (including gliomas) is feasible (PMID 38227143) and one showing that multiplex ligation-dependent probe amplification (MLPA) in CSF ctDNA is possible for diagnosis of gliomas (PMID 36875626) are important papers that should be cited.

Author Response

Reviewer 1:

Author's Reply to the Review Report (Reviewer 1)

Comments and Suggestions for Authors

Dr. Seyhan writes a comprehensive review of where we stand in terms of liquid biopsy in glioblastoma. The paper is generally well-written. The following changes should be considered.

Response to reviewer 1: The author thanks the reviewer for taking the time to review this manuscript and for the comments which were very helpful in revising the manuscript to convey the message more clearly. Below is a point-by-point response to each comment. The corresponding revisions/corrections highlighted/in track changes in the re-submitted files.

  1. In the introduction section, the authors cite a paper showing that mOS of GBM is about 8 months (line 52), and a different papers showing the mOS to be less than 15 months. The former paper is probably "real-world" data and the latter are clinical trials, but the two seem to contradict each other, so a reader not familiar with GBMs may be confused. Please clarify.

Author’s response: The author thanks the reviewer for the important critique and helpful comments.

The median OS of GBM patients has been revised to 15 months throughout the text in line with the current literature.

“Median survival time of about 8 months” has been deleted and only "a median survival of less than 15 months" has been used throughout the text.

  1. Many of the paragraphs in the paper are only one sentence. This may be a matter of style, but paragraphs should be multiple sentences which are related to each other. Please correct.

Author’s response: The author thanks the reviewer for the critique and helpful comments.

Many of the long sentences were constructed to retain the intended meaning and the description of te subject matter without breaks. However, in agreement with the reviewer’s comments, some of the long sentences were broken down into multiple sentences while maintaining the intended meaning.

  1. The following papers, one showing that liquid biopsy of cyst fluid in cystic brain tumors (including gliomas) is feasible (PMID38227143) and one showing that multiplex ligation-dependent probe amplification (MLPA) in CSF ctDNA is possible for diagnosis of gliomas (PMID 36875626) are important papers that should be cited.

Author’s response: The author thanks the reviewer for bringing these important papers to my attention.

The mentioned papers (PMID38227143 and PMID 36875626) are now cited in the appropriate places in the text. [1] [2]

Page 10, line 318 - For example, a recent study has demonstrated that cfDNA from cyst fluid in cystic brain tumors is a reliable alternative to tumor DNA for diagnosing brain tumors [83]. PMID38227143

Page 13, line 430 - In addition, a recent study has demonstrated that copy number analysis can be effectively performed with multiplex ligation-dependent probe amplification of cfDNA in CSF from patients with adult diffuse glioma [115]. PMID 36875626

On J, et al: Reliable detection of genetic alterations in cyst fluid DNA for the diagnosis of brain tumors. J Neurooncol 2024, 166:273-282. (PMID38227143)

Otsuji R, et al: Liquid biopsy with multiplex ligation-dependent probe amplification targeting cell-free tumor DNA in cerebrospinal fluid from patients with adult diffuse glioma. Neurooncol Adv 2023, 5:vdac178. (PMID 36875626)

Reviewer 2 Report

Comments and Suggestions for Authors

Journal IJMS (ISSN 1422-0067)

Manuscript ID ijms-3080100

Title Circulating liquid biopsy biomarkers in glioblastoma: Advances and challenges

Authors Attila A Seyhan *

Seyhan A. A. prepared an outstanding review on the utility of different liquid biopsy markers in prognostication/potential detection of the glioblastoma type of cancers. The subjects covered are presented in a nicely organized sentence, the cited literature is appropriate and up to date. The presentation of clinical trials involving different biomarkers is very relevant and well organized. Figures are of high quality and very informative.

The contrast of the use of the standard of care vs. the use of liquid biopsy markers in the context of precision medicine is of high value for the reader. This review is well constructed for the audience who is well-versed in precision medicine using liquid biopsy markers and for those who are novice in the field is new. There are only minor edits suggested to the manuscript. Please see below. Overall, please publish this manuscript.

1.    Please explain acronym IDH on page 2.

2.    Page 7: As highlighted in te recent literature [75, 76] “The” is misspelled.

3.    Page 7: While most of these biomarkers have a short half-life (~3 h) and.. I think better to specify that this pertains to nucleic acids. CTC can persist in blood longer than 3 h.

4.    Page 10 “Table 10 illustrates examples of several prospective clinical studies.” This should be Table 1. On that note, please correct the order of all the tables or the numbering of the tables, so they appear in a consecutive order. For example: Page 12: Should be Table 3 not 6, and others in the manuscript.

Author Response

Reviewer 2:

Author's Reply to the Review Report (Reviewer 2)

Comments and Suggestions for Authors

Seyhan A. A. prepared an outstanding review on the utility of different liquid biopsy markers in prognostication/potential detectionof the glioblastoma type of cancers. The subjects covered are presented in a nicely organized sentence, the cited literature is appropriate and up to date. The presentation of clinical trials involving different biomarkers is very relevant and well organized. Figures are of high quality and very informative.

The contrast of the use of the standard of care vs. the use of liquid biopsy markers in the context of precision medicine is of high value for the reader. This review is well constructed for the audience who is well-versed in precision medicine using liquid biopsy markers and for those who are novice in the field is new. There are only minor edits suggested to the manuscript. Please see below. Overall, please publish this manuscript.

Response to Reviewer 2: 

The author thanks the reviewer for taking the time to review this manuscript and for the comments which were very helpful in revising the manuscript to convey the message more clearly.

Below is a point-by-point response to each comment. The corresponding revisions/corrections highlighted/in track changes in the re-submitted files.

Author's Reply to the Review Report (Reviewer 2)

  1. Please explain acronym IDH on page 2.

Author’s response: The author thanks the reviewer for the important critique and helpful comments.

Corrected. The abbreviation of IDH was typed out when it was introduced the first time as “isocitrate dehydrogenase” (IDH)-wild type. Page 2, line 74.

  1. Page 7: As highlighted in te recent literature [75, 76] “The” is misspelled.

Author’s response: Corrected. “te” changed to “the”. Page 8, line 284.

  1. Page 7: While most of these biomarkers have a short half-life (~3 h) and. I think better to specify that this pertains to nucleic acids.CTC can persist in the blood longer than 3 h.

Author’s response: The half-life of ctDNA and other biomarkers such as miRNAs, CTCs, EVs, exosomes, proteins, and metabolites was revised according to recent literature: Page 22, line 926 and Table 10.

  1. Page 10 “Table 10 illustrates examples of several prospectiveclinical studies.” This should be Table 1. On that note, please correct the order of all the tables or the numbering of the tables, so they appear in a consecutive order. For example: Page 12: Shouldbe Table 3 not 6, and others in the manuscript.

Author’s response: The order and numbering of all tables have been revised and corrected. They now appear in consecutive order."

Reviewer 3 Report

Comments and Suggestions for Authors

The author presents an up-to-date review of the various categories of molecules that can be assessed via liquid biopsy for glioblastoma (GBM) in adults. Taken together, the manuscript is well-written, comprehensive and well-organized. However, there are major, specific and minor concerns that should be addressed as follows:

A. Major concerns

1. Overall, the first 3 sections (pages 1-6) pertaining to clinical diagnosis and management of adult glioblastoma (GBM) are not necessary and significantly detract from the manuscript which is currently 61 pages including figures, tables and references. Readers interested in a timely review of the clinical management of GBM have many excellent options recently published in the literature. Please refer to many examples of specific concerns in the next section of this review.

2. One-sentence paragraphs are not accepted with rare exception. In several parts of the manuscript there are runs of one-sentence paragraphs. Please refer to many examples of specific concerns in the next section of this review.

3. The author implies, perhaps unintentionally, in several places of the manuscript that a maximal safe resection or even a biopsy is dangerous. On the contrary, the extent of resection correlates with improved PFS and sometimes OS in limited retrospective studies in both paediatric and adulot high-grade gliomas, including GBM. Liquid biopsy is reasonable to be obtained at various times in the patient journey, including at diagnosis, but it cannot replace MRI imaging or biopsy/maximal safe resection, depending upon the neuroanatomic location of the tumour.

B. Specific concerns

1. Abstract, page 1, lines 15-16: Delete "both of which have limitations".

2. Section 1, Introduction, page 2, line 52: The statement "median survival time of about 8 months" contradicts line 66: "a median survival of less than 15 months". Please be consistent.

3. Lines 62-64: The viral hypothesis regarding the etiology of GBM is not yet established and is considered by many authorities as controversial. This statement should be revised or deleted, accordingly.

4. Lines 78-80: This statement should be revised as per WHO CNS5 (2021). This is a one sentence paragraph (also lines 97-98 and 104-106).

5. Lines 86-90: The classical histological features of GBM and more common molecular features should be listed in separate sentences and not mixed together.

6. It is unclear why a review article on liquid biopsy of GBM is including discussion of immune therapy, including immune checkpoint inhibitors. There are many authoritative reviews on this topic recently published. Also, its inclusion in this section of the Introduction seems out of place. Hence, deletion of lines 107-110 is highly recommended.

7. Lines 113-116 are incorrect as written and out-of-place. Surgery (for biopsy or maximal safe resection) are necessary for diagnosis, molecular sub-classification and treatment assignment. Stating that they are risky will confuse the reader who is not a neurosurgeon, radiation or neuro-oncologist. How can "treatment-related changes" be referred to at tumour diagnosis?

8. Lines 125-127 (a 1-sentence paragraph) should be placed much earlier in the article.

9. Section 2, Current approaches for the diagnosis of GBM, pages 3-5

-Lines 137-140: Liquid biopsy doesn't diminish the role of initial tumour biopsy or maximal safe resection as determined by a neurosurgeon. Furthermore, as stated above, there may be prognostic benefits from the extent of resection. Hence, this 1-sentence paragraph should be revised, accordingly.

10. Section 3, Current standard of care for treating GBM (pages 5-7)

-The sections on immunotherapy, including immune checkpoint inhibitors and tumour vaccines, are not of direct relevance to the topic of liquid biopsy, and take up over one page of text. It is highly recommended that lines 226-275 either be deleted or substantially reduced. As stated elsewhere, there are several recent reviews on this topic published in this journal and others more focused to the field of neuro-oncology.

11. Section 5, Liquid biopsies in cancer

-Lines 335-340: In paediatrics, especially for medulloblastoma and other embryonal tumours, CSF sampling (lumbar, postoperative) is a mandatory part of staging and is considered as minimally invasive as they are frequently performed under conscious sedation or general anaesthetic. Furthermore, there is emerging evidence for diffuse midline gliomas (DMG H3K27 altered), especially of the pons, that CSF is more reliable than blood, serum or plasma. There was an excellent paper by Anthony Liu and colleagues from the St Jude Children's Research Hospital on the utility of CSF as liquid biopsy for medulloblastoma in Cancer Cell (2021).

12. Section 7, Circulating miRNA profiling as potential biomarkers

-Lines 488-492: Please comment about the specificity. From the reviewer's recollection, specificity was below the stated sensitivity of 84%.

13. Section 8, CTCs as potential biomarkers

-Line 642-643: EGFR amplification is only found in a subset of GBM and this is even more so for the EGFRvIII variant. Please provide more context for this statement as it pertains to CTCs.

14. Section 13, Conclusions

-Line 1049 is problematic as written. Surveillance MRIs are essential for following patients after initial diagnosis and therapy. Cost-effectiveness is relative to alternatives, which to date have yet to be established. Maximal safe surgical resection is part of the standard of care. However, the author has repeated commented on the complications of therapy, including in this instance "neurological deficits". Unless there is literature and statistics to back up this statement it should be rewritten or deleted.

15. Tables

-Table 2: How was this table organized? It might be easier for the reader to find specific miRs if they were organized in numerical order. If not, then justify the current organization.

-Table 4 is missing.

C. Minor concerns

1. Introduction

-Page 2, line 47: Delete "multiforme". This term has not been used since the WHO CNS4 (2016).

-Line 51: Add "CNS" prior to "tumors".

2. Current approaches for the diagnosis of GBM (pages 3-5)

-Line 136: What is meant by the phrase "characterize its properties" from the perspective of tumour diagnosis?

-Line 144: Spell out the first use of all abbreviations; in this case "IHC".

-Lines  170-183: It is unclear why this list of terms is included for the reader. As presented, it serves no purpose and it is highly recommended that it is deleted or made into a supplementary table (i.e. not part of the main manuscript).

3. Current standard of care for treating GBM (pages 5-7)

-This is a review article on liquid biopsy for GBM; as such, only a brief cursory review of the current management is recommended. Certainly, there is no place for drug dosing and dosing intervals. If interested, the reader can refer elsewhere to articles focused on management. Please reduce the first 6 paragraphs of this section substantially so as not to detract from the purpose of this comprehensive review article.

-Lines 203-205: This is the third time the median survival of GBM patients has been stated; this is repetitive.

-Lines 218-220: This was presented earlier and is repetitive.

-Lines 221-225: This a 1-sentence paragraph, was presented earlier and is repetitive.

4. Section 5, Liquid biopsies in cancer

-Line 300: Correctly spell "the".

-Figure 2. Under "Liquid biopsy, disadvantages". Place a period after "evaluation".

-Lines 300-313: There are five 1-sentence paragraphs in a row; convert to one or two multi-sentence paragraphs.

-Lines 341-348: There are 3 1-sentence paragraphs in a row. Consolidate into one paragraph or incorporate into the preceding or subsequent paragraphs, respectively.

-Figure 3: Under miRNA: Correct the spelling of "yield" and "high" throughput.

5. Section 6, ctDNA profiling as potential biomarkers for GBM.

-Lines 403-411: There are 3 consecutive 1-sentence paragraphs.

-Lines 416-418: This is a 1-sentence paragraph.

-Lines 435-438: This is a 1-sentence paragraph.

-Lines 446-448: This is a 1-sentence paragraph.

-Lines 458-460: This is a 1-sentence paragraph.

6. Section 7, Circulating miRNA profiling as potential biomarkers

-Lines 513-515, 524-525, 526-529 and 543-545 are all 1-sentence paragraphs.

-Line 528: Spell "from" correctly.

-Line 530: Delete "in" prior to "recent".

-Line 558: Change "correlate" to "correlates".

-Lines 568-570 and 578-580 are 1-sentence paragraphs.

7. Section 8, CTCs as potential biomarkers.

-Lines 594-596 is a 1-sentence paragraph, as are lines 616-618, lines 644-645, lines 652-654 and 669-672.

8. Section 9, Circulating protein profiling as potential biomarkers.

-Lines 706-708, 709-713, 714-715, 728-731, 732-735, 742-744,  and 773-775 are all 1-sentence paragraphs.

9. Section 10, circulating matabolomic and lipidomic profiling as potential biomarkers

-Lines 777-779, 786-789, 790-792, 793-797, 803-808, 809-811 are all 1-sentence paragraphs.

-Line 794: This sentence may be confusing to the reader. GBM tumours are high grade gliomas so the term "higher-grade brain tumour" is not necessary.

-Line 804: Change to "lipidomics".

-Line 817: Change "treatments" to "treatment".

-Lines 812-813, 814-815, 826-830, 831-834, and 835-837 are all 1-sentence paragraphs.

10. Section 11, Circulating extracellular vesicle (EV) and exosome profiling as potential biomarkers

-Lines 840-862: There are nine consecutive 1-sentence paragraphs.

-Line 854: What is meant by "stable" in the context of this section of the manuscript?"

-Lines 875-878, 879-880, 890-891, 892-894, 895-897 are all 1-sentence paragraphs.

11. Section 12, Challenges and future perspectives

-Lines 899-901, 902-903, 910-912 are all 1-sentence paragraphs.

-Line 904: Change "Traditional" to "traditional".

-Lines 917-919, 920-922, 923-925, 942-945 are all 1-sentence paragraphs.

-Line 938: Change to "detection".

-Line 950: Add "potentially" prior to "carries more risk".

-Lines 986-990, 991-993, 994-997, and 998-1002 are all 1-sentence paragraphs.

-Lines 1028-1031, 1032-1035, and 1036-1038 are all 1-sentence paragraphs.

12. Section 13, Conclusions

-Lines 1050-1053, 1064-1066, 1067-1069, and 1070-1075 are all 1-sentence paragraphs.

-Lines 1076-1078, 1079-1083, 1084-1089, and 1090-1091 are all 1-sentence paragraphs.

Comments on the Quality of English Language

Overall, the written English language is correct. However, the dozens of 1-sentence paragraphs would not be accepted at most journals and in the majority of textbooks.

Author Response

Reviewer 3:

Author's Reply to the Review Report (Reviewer 3)

Comments and Suggestions for Authors

The author presents an up-to-date review of the various categories of molecules that can be assessed via liquid biopsy for glioblastoma (GBM) in adults. Taken together, the manuscript is well-written, comprehensive and well-organized. However, there are major, specific and minor concerns that should be addressed as follows:

  1. Major concerns
  2. Overall, the first 3 sections (pages 1-6) pertaining to clinical diagnosis and management of adult glioblastoma (GBM) are not necessary and significantly detract from the manuscript which is currently 61 pages including figures, tables and references. Readers interested in a timely review of the clinical management of GBM have many excellent options recently published in the literature. Please refer to many examples of specific concerns in the next section of this review.

Author’s response: The author expresses sincere gratitude to the reviewer for the time and valuable comments provided.

The mentioned sections have been either shortened, updated with recent literature, or removed. Nevertheless, the author strongly believes that an introduction to these sections is crucial. It offers essential context for newcomers to the field and enhances the paper's informativeness for both experts and novices. Thus, maintaining these introductory elements ensures the manuscript serves a broader audience effectively.

  1. One-sentence paragraphs are not accepted with rare exception. In several parts of the manuscript there are runs of one-sentence paragraphs. Please refer to many examples of specific concerns in the next section of this review.

Author’s response: The author thanks the reviewer for the critique and helpful comments.

The one-sentence paragraphs were consolidated into one or two multi-sentence paragraphs if the theme is the same or incorporated into the preceding or subsequent paragraphs, respectively.

  1. The author implies, perhaps unintentionally, in several places of the manuscript that a maximal safe resection or even a biopsy is dangerous. On the contrary, the extent of resection correlates with improved PFS and sometimes OS in limited retrospective studies in both paediatric and adulot high-grade gliomas, including GBM. Liquid biopsy is reasonable to be obtained at various times in the patient journey, including at diagnosis, but it cannot replace MRI imaging or biopsy/maximal safe resection, depending upon the neuroanatomic location of the tumour.

Author’s response: The author thanks the reviewer for the critique and helpful comments.

In agreement with the reviewer’s comments, the text has been revised such that the mention of “maximal safe resection or even a biopsy is dangerous”  has been modified and “dangerous” has been eliminated.

  1. Specific concerns
  2. Abstract, page 1, lines 15-16: Delete "both of which have limitations".

Author’s response: Corrected accordingly.

  1. Section 1, Introduction, page 2, line 52: The statement "median survival time of about 8 months" contradicts line 66: "a median survival of less than 15 months". Please be consistent.

Author’s response: Corrected.

“Median survival time of about 8 months” has been deleted and only "a median survival of less than 15 months" has been used throughout the text.

  1. Lines 62-64: The viral hypothesis regarding the etiology of GBM is not yet established and is considered by many authorities as controversial. This statement should be revised or deleted, accordingly.

Author’s response: Corrected accordingly and included the following sentence following the viral hypothesis etiology: “However, the viral hypothesis regarding the etiology of GBM is not yet well established and is considered as controversial by many authorities.

  1. Lines 78-80: This statement should be revised as per WHO CNS5 (2021). This is a one sentence paragraph (also lines 97-98 and 104-106).

Author’s response: Revised accordingly.

  1. Lines 86-90: The classical histological features of GBM and more common molecular features should be listed in separate sentences and not mixed together.

Author’s response: Revised accordingly.

  1. It is unclear why a review article on liquid biopsy of GBM is including discussion of immune therapy, including immune checkpoint inhibitors. There are many authoritative reviews on this topic recently published. Also, its inclusion in this section of the Introduction seems out of place. Hence, deletion of lines 107-110 is highly recommended.

Author’s response: Revised accordingly.

It is the author’s opinion that a cursory introduction to emerging therapies such as immunotherapy make this paper more informative for the audience who is experts in the field as well as for those who are new to the field.

  1. Lines 113-116 are incorrect as written and out-of-place. Surgery (for biopsy or maximal safe resection) are necessary for diagnosis, molecular sub-classification and treatment assignment. Stating that they are risky will confuse the reader who is not a neurosurgeon, radiation or neuro-oncologist. How can "treatment-related changes" be referred to at tumour diagnosis?

Author’s response: Revised accordingly.

  1. Lines 125-127 (a 1-sentence paragraph) should be placed much earlier in the article.

Author’s response: The mentioned sentence is combined with the previous paragraph. Because a similar sentence was already used in the abstract, keeping this sentence as a closing statement in the introduction is preferred.

  1. Section 2, Current approaches for the diagnosis of GBM, pages 3-5

-Lines 137-140: Liquid biopsy doesn't diminish the role of initial tumour biopsy or maximal safe resection as determined by a neurosurgeon. Furthermore, as stated above, there may be prognostic benefits from the extent of resection. Hence, this 1-sentence paragraph should be revised, accordingly.

Author’s response: Revised accordingly.

  1. Section 3, Current standard of care for treating GBM (pages 5-7)

-The sections on immunotherapy, including immune checkpoint inhibitors and tumour vaccines, are not of direct relevance to the topic of liquid biopsy, and take up over one page of text. It is highly recommended that lines 226-275 either be deleted or substantially reduced. As stated elsewhere, there are several recent reviews on this topic published in this journal and others more focused to the field of neuro-oncology.

Author’s response: Revised accordingly.

As noted above, it is the author’s opinion that some introduction to emerging innovative therapies for GBM make this paper more informative for the wider audience who is new to the field and experts in the field.

  1. Section 5, Liquid biopsies in cancer

-Lines 335-340: In paediatrics, especially for medulloblastoma and other embryonal tumours, CSF sampling (lumbar, postoperative) is a mandatory part of staging and is considered as minimally invasive as they are frequently performed under conscious sedation or general anaesthetic. Furthermore, there is emerging evidence for diffuse midline gliomas (DMG H3K27 altered), especially of the pons, that CSF is more reliable than blood, serum or plasma. There was an excellent paper by Anthony Liu and colleagues from the St Jude Children's Research Hospital on the utility of CSF as liquid biopsy for medulloblastoma in Cancer Cell (2021).

Author’s response: Revised accordingly and included this information in this section.

  1. Section 7, Circulating miRNA profiling as potential biomarkers

-Lines 488-492: Please comment about the specificity. From the reviewer's recollection, specificity was below the stated sensitivity of 84%.

Author’s response: Revised accordingly.

  1. Section 8, CTCs as potential biomarkers

-Line 642-643: EGFR amplification is only found in a subset of GBM, and this is even more so for the EGFRvIII variant. Please provide more context for this statement as it pertains to CTCs.

Author’s response: Revised accordingly.

  1. Section 13, Conclusions

-Line 1049 is problematic as written. Surveillance MRIs are essential for following patients after initial diagnosis and therapy. Cost-effectiveness is relative to alternatives, which to date have yet to be established. Maximal safe surgical resection is part of the standard of care. However, the author has repeated commented on the complications of therapy, including in this instance "neurological deficits". Unless there is literature and statistics to back up this statement it should be rewritten or deleted.

Author’s response: Revised accordingly.

 The original sentence “This approach holds particular importance for CNS tumors, as traditional MRI scans are costly, and surgical resection can lead to neurological deficits.” is modified as follows:

“This approach holds particular importance for CNS tumors complementing the traditional MRI scans for monitoring patients after initial diagnosis and therapy.”

  1. Tables

-Table 2: How was this table organized? It might be easier for the reader to find specific miRs if they were organized in numerical order. If not, then justify the current organization.

Author’s response: Table 4 is now organized in numerical order.

-Table 4 is missing.

Author’s response: Corrected. Tables were renumbered.

  1. Minor concerns
  2. Introduction

-Page 2, line 47: Delete "multiforme". This term has not been used since the WHO CNS4 (2016).

Author’s response: Corrected.

-Line 51: Add "CNS" prior to "tumors".

Author’s response: Corrected.

  1. Current approaches for the diagnosis of GBM (pages 3-5)

-Line 136: What is meant by the phrase "characterize its properties" from the perspective of tumour diagnosis?

Author’s response: Revised as “…characterize its pathological, genetic, genomic, transcriptomic, proteomic, and other molecular properties."

-Line 144: Spell out the first use of all abbreviations; in this case "IHC".

Author’s response: Corrected.

-Lines  170-183: It is unclear why this list of terms is included for the reader. As presented, it serves no purpose and it is highly recommended that it is deleted or made into a supplementary table (i.e. not part of the main manuscript).

Author’s response: Deleted.

  1. Current standard of care for treating GBM (pages 5-7)

-This is a review article on liquid biopsy for GBM; as such, only a brief cursory review of the current management is recommended. Certainly, there is no place for drug dosing and dosing intervals. If interested, the reader can refer elsewhere to articles focused on management. Please reduce the first 6 paragraphs of this section substantially so as not to detract from the purpose of this comprehensive review article.

Author’s response: Revised.

-Lines 203-205: This is the third time the median survival of GBM patients has been stated; this is repetitive.

Author’s response: Deleted.

-Lines 218-220: This was presented earlier and is repetitive.

Author’s response: Deleted.

-Lines 221-225: This a 1-sentence paragraph, was presented earlier and is repetitive.

Author’s response: Revised.

  1. Section 5, Liquid biopsies in cancer

-Line 300: Correctly spell "the".

Author’s response: Corrected.

-Figure 2. Under "Liquid biopsy, disadvantages". Place a period after "evaluation".

-Lines 300-313: There are five 1-sentence paragraphs in a row; convert to one or two multi-sentence paragraphs.

Author’s response: Corrected.

-Lines 341-348: There are 3 1-sentence paragraphs in a row. Consolidate into one paragraph or incorporate into the preceding or subsequent paragraphs, respectively.

Author’s response: Corrected.

-Figure 3: Under miRNA: Correct the spelling of "yield" and "high" throughput.

  1. Section 6, ctDNA profiling as potential biomarkers for GBM.

-Lines 403-411: There are 3 consecutive 1-sentence paragraphs.

Author’s response: Corrected.

-Lines 416-418: This is a 1-sentence paragraph.

Author’s response: Corrected.

-Lines 435-438: This is a 1-sentence paragraph.

Author’s response: Corrected.

-Lines 446-448: This is a 1-sentence paragraph.

Author’s response: Corrected.

-Lines 458-460: This is a 1-sentence paragraph.

Author’s response: Corrected.

  1. Section 7, Circulating miRNA profiling as potential biomarkers

-Lines 513-515, 524-525, 526-529 and 543-545 are all 1-sentence paragraphs.

Author’s response: Corrected.

-Line 528: Spell "from" correctly.

Author’s response: Corrected.

-Line 530: Delete "in" prior to "recent".

Author’s response: Corrected.

-Line 558: Change "correlate" to "correlates".

Author’s response: Corrected.

-Lines 568-570 and 578-580 are 1-sentence paragraphs.

Author’s response: Corrected.

  1. Section 8, CTCs as potential biomarkers.

-Lines 594-596 is a 1-sentence paragraph, as are lines 616-618, lines 644-645, lines 652-654 and 669-672.

  1. Section 9, Circulating protein profiling as potential biomarkers.

-Lines 706-708, 709-713, 714-715, 728-731, 732-735, 742-744,  and 773-775 are all 1-sentence paragraphs.

Author’s response: Corrected.

  1. Section 10, circulating matabolomic and lipidomic profiling as potential biomarkers

-Lines 777-779, 786-789, 790-792, 793-797, 803-808, 809-811 are all 1-sentence paragraphs.

Author’s response: Corrected.

-Line 799: This sentence may be confusing to the reader. GBM tumours are high grade gliomas so the term "higher-grade brain tumour" is not necessary.

-Line 804: Change to "lipidomics".

Author’s response: Corrected.

-Line 817: Change "treatments" to "treatment".

Author’s response: Corrected.

-Lines 812-813, 814-815, 826-830, 831-834, and 835-837 are all 1-sentence paragraphs.

Author’s response: Corrected.

  1. Section 11, Circulating extracellular vesicle (EV) and exosome profiling as potential biomarkers

-Lines 840-862: There are nine consecutive 1-sentence paragraphs.

Author’s response: Corrected.

-Line 854: What is meant by "stable" in the context of this section of the manuscript?"

Author’s response: Revised as follows.

“Furthermore, EVs and exosomes contain RNAs such as mRNAs and miRNAs, DNA, proteins, and lipids, all shielded from enzymatic degradation [47]. This composition offers a more accurate representation of biological processes compared to ctDNA or other biomarkers.“

-Lines 875-878, 879-880, 890-891, 892-894, 895-897 are all 1-sentence paragraphs.

Author’s response: Corrected.

  1. Section 12, Challenges and future perspectives

-Lines 899-901, 902-903, 910-912 are all 1-sentence paragraphs.

Author’s response: Corrected.

-Line 904: Change "Traditional" to "traditional".

Author’s response: Corrected.

-Lines 917-919, 920-922, 923-925, 942-945 are all 1-sentence paragraphs.

Author’s response: Corrected.

-Line 938: Change to "detection".

Author’s response: Corrected.

-Line 950: Add "potentially" prior to "carries more risk".

Author’s response: Corrected.

-Lines 986-990, 991-993, 994-997, and 998-1002 are all 1-sentence paragraphs.

Author’s response: Corrected.

-Lines 1028-1031, 1032-1035, and 1036-1038 are all 1-sentence paragraphs.

Author’s response: Corrected.

  1. Section 13, Conclusions

-Lines 1050-1053, 1064-1066, 1067-1069, and 1070-1075 are all 1-sentence paragraphs.

-Lines 1076-1078, 1079-1083, 1084-1089, and 1090-1091 are all 1-sentence paragraphs.

Comments on the Quality of English Language

Overall, the written English language is correct. However, the dozens of 1-sentence paragraphs would not be accepted at most journals and in the majority of textbooks.

Round 2

Reviewer 3 Report

Comments and Suggestions for Authors

The author has comprehensively addressed all of the substantive as well as minor concerns raised by the initial review.

However, one minor revision is necessary (line 199, revised manuscript). The nitrosourea used for adult GBM at disease recurrence and for adult-type paediatric HGG at diagnosis, combined with temozolomide, is lomustine (CCNU), given orally. Carmustine (BCNU) is only delivered using drug-impregnated wafers placed at the time or initial or reoperation in the tumour cavity (Gliadel is the trade name). However, academic neurosurgeons have not favoured the use of carmustine wafers. Hence, this part of the introduction requires revision to ensure clinical accuracy.

Author Response

Reviewer 3:

Comments and Suggestions for Authors

The author has comprehensively addressed all of the substantiveas well as minor concerns raised by the initial review.

However, one minor revision is necessary (line 199, revised manuscript). The nitrosourea used for adult GBM at disease recurrence and for adult-type paediatric HGG at diagnosis, combined with temozolomide, is lomustine (CCNU), given orally.Carmustine (BCNU) is only delivered using drug-impregnated wafers placed at the time or initial or reoperation in the tumourcavity (Gliadel is the trade name). However, academic neurosurgeons have not favoured the use of carmustine wafers. Hence, this part of the introduction requires revision to ensure clinical accuracy.

Response to reviewer 3: The author thanks the reviewer for taking the time to review this manuscript and for the comments which were very helpful in revising the manuscript to convey the message more clearly. Below is a point-by-point response to each comment. The corresponding revisions/corrections highlighted/in track changes in the re-submitted files.

Author’s response: The author thanks the reviewer for the critique and helpful comments.

In agreement with the reviewer’s comments, the text has been revised.

Submission Date: 12 June 2024

Date of this review: 16 Jul 2024 08:48:13
